# Seroprevalence and SARS-CoV-2 invasion in general populations: A scoping review over the first year of the pandemic

**Clémentine Metzger**[1], **Taylor Leroy**[1], **Agathe Bochnakian**[1], **Hélène Jeulin**[2,3], **Anne Gegout-Petit**[4], **Karine Legrand**[1], **Evelyne Schvoerer**[2,3], **Francis Guillemin**[1]*

**1** CHRU -Nancy, INSERM, Université de Lorraine, CIC Epidémiologie clinique, F-54000, Nancy, France, **2** Université de Lorraine, CNRS, LCPME, F-54000, Nancy, France, **3** Laboratoire de Virologie, CHRU de Nancy Brabois, F-54500, Nancy, France, **4** Université de Lorraine, CNRS, Inria, IECL, F-54000, Nancy, France

* francis.guillemin@chru-nancy.fr

**Data Availability Statement:** Toutes les données pertinentes se trouvent dans le manuscrit et ses fichiers d'informations à l'appui All relevant data is

## Abstract

Since the beginning of the COVID-19 pandemic, counting infected people has underestimated asymptomatic cases. This literature scoping review assessed the seroprevalence progression in general populations worldwide over the first year of the pandemic. Seroprevalence studies were searched in PubMed, Web of Science and medRxiv databases up to early April 2021. Inclusion criteria were a general population of all ages or blood donors as a proxy. All articles were screened for the title and abstract by two readers, and data were extracted from selected articles. Discrepancies were resolved with a third reader. From 139 articles (including 6 reviews), the seroprevalence estimated in 41 countries ranged from 0 to 69%, with a heterogenous increase over time and continents, unevenly distributed among countries (differences up to 69%) and sometimes among regions within a country (up to 10%). The seroprevalence of asymptomatic cases ranged from 0% to 31.5%. Seropositivity risk factors included low income, low education, low smoking frequency, deprived area residency, high number of children, densely populated centres, and presence of a case in a household. This review of seroprevalence studies over the first year of the pandemic documented the progression of this virus across the world in time and space and the risk factors that influenced its spread.

## Introduction

The coronavirus disease outbreak, caused by severe acute respiratory syndrome coronavirus 2 (SARS-CoV-2), was first reported in Wuhan in December 2019 and spread rapidly to other parts of the world. On April 30, 2021, COVID-19 accounted for 150,000,000 confirmed cases worldwide, more than 3,000,000 deaths and about 87,000,000 recoveries, representing the deadliest pandemic in decades [1]. To contain the spread of the virus, daily counts of laboratory-confirmed cases and deaths have been published in real time.

in the manuscript and its supporting information files.

**Funding:** The study was funded by Metropole du Grand (https://www.grandnancy.eu/accueil/). The funder had no role in study design, data collection and analysis, decision to publish, or preparation of the manuscript.

**Competing interests:** The authors have declared that no competing interests exist.

The emergence of this new virus has resulted in an important early warning plan, and the WHO has tried to understand its modes of transmission, severity, characteristics and risk factors for infection. This alert plan was targeted to manage this epidemic, which became a pandemic barely 2 months after its appearance [2]. The plan includes refining case referrals, reinforcing surveillance, and defining the main epidemiological characteristics of COVID-19 to help understand the spread, severity, spectrum of the disease and impact on the community and to provide guidance on the application of countermeasures such as case isolation and contact tracing. Daily counts of confirmed COVID-19 cases and deaths alone provide incomplete information on the relative abundance of epidemiological compartments of a susceptible infected, recovered or deceased population. Therefore, the WHO has recommended repeatedly carrying out seroprevalence surveys in multiple geographical settings [2].

Several serological surveys of SARS-CoV-2 have been performed, and others are ongoing since the beginning of the COVID-19 pandemic, finding variable seroprevalence in different countries, sometimes even among regions of the same country [3,4].

The purpose of this literature scoping review was to summarise and map the results of the seroprevalence studies according to the time since the onset of the pandemic and geographical region and to identify risk factors.

## Methods

We performed a literature review of seroprevalence studies conducted in different populations since the onset of the pandemic up to April 10, 2021, searching PubMed, Web of Science and medRxiv databases. We searched references of citations for reviews, which allowed for additional references to be added.

### Eligibility

We included original articles and reviews written in English language that reported data for the general population living in a defined geographical area. We considered a general population as a population-based sample, a described general population sample, or the inclusion of men and women of unselected categories of all ages in the population, preferably (but not exclusively) obtained by a random sampling technique from a large population (survey or database). We also included blood donors as a proxy for the general population. We excluded studies of health care workers, people attending a clinic or a hospital, a professional branch (industry, factory, farmers, university) and a particular population (students, nursing home) as well as modelling studies.

### Search strategy

The strategy consisted of searching PubMed and medRxiv with "(Covid OR SARS-CoV-2) AND seroprevalence" and Web of Science with "covid AND seroprevalence". Articles were selected by reading titles and abstracts. All article titles were first read by two readers (CM, AB), then by a third one (FG or TL), which allowed for discussion in case of discrepancies after abstract reading to obtain consensus. Full-article reading allowed for the final selection of relevant studies and data extraction.

### Data extraction

Data extraction was conducted by three authors who used a standardized form. The data collected, when available, were location; selection criteria; type of population; sample size and method; seroprevalence; serological test and type of Ig antibody tested; time period of the

study; presence and type of symptoms; risk factors such as age, sex, ethnicity and origin; local medical resources; and social class.

## Results

### Study characteristics

The search strategy yielded 742 articles from PubMed, 281 from Web of Science and 1141 from medRxiv. The selection of relevant articles, after removal of duplicates, yielded 139 articles, including 6 reviews and meta-analysis [5–10], published from January 1, 2020 to April 10, 2021 (Fig 1). Article references described in this section refer to those in Table 1. Of the 133 original studies, 31 were of blood donors. In 64 studies, the sample included all ages; in 40, it included people $\geq$ 18 years old; and a few studied people < age 18 years (n = 17) or 10 years (n = 12). Some studies identified co-morbidities or asymptomatic cases. The samples ranged from 194 to 10,294,728 participants.

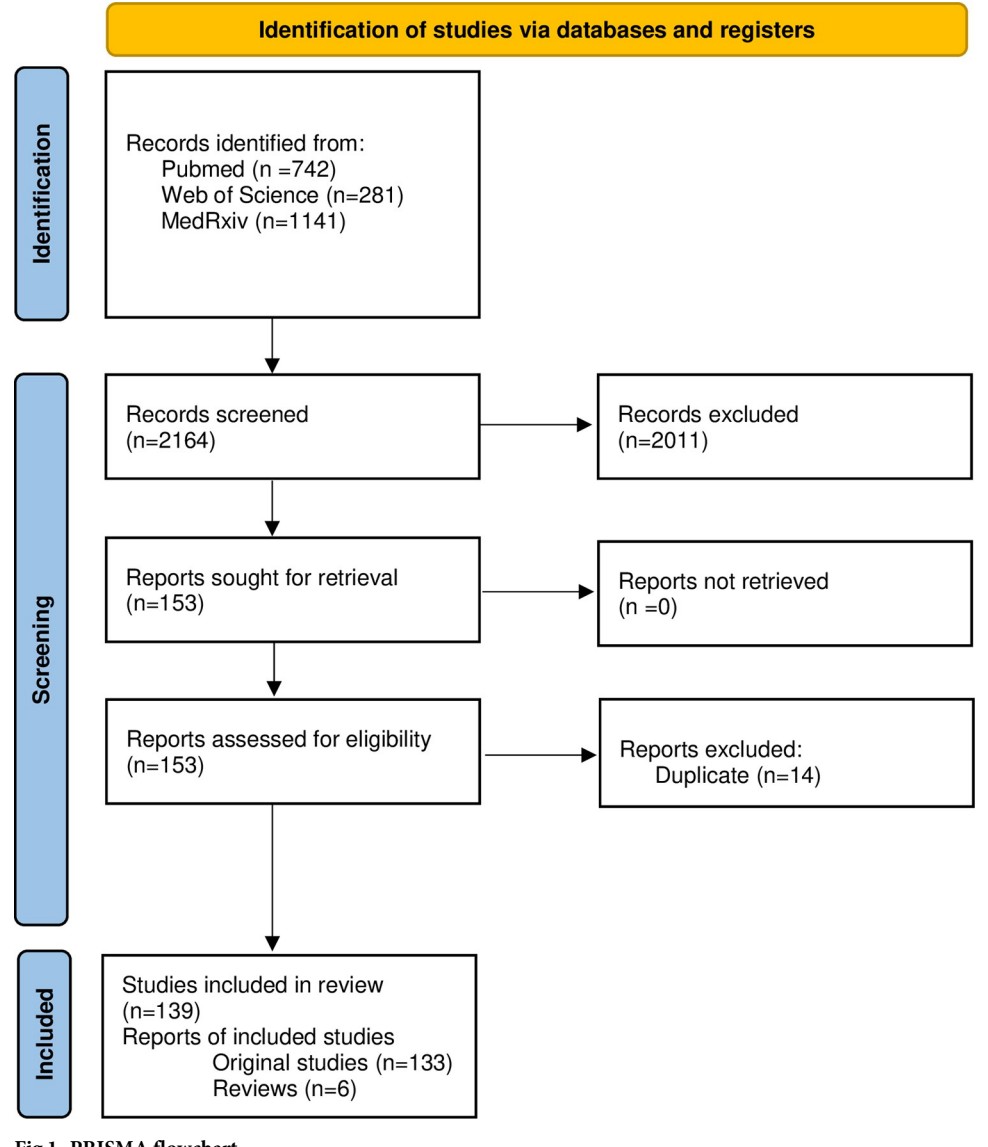

**Fig 1. PRISMA flowchart.**

**Table 1. Characteristics of country, methods, population of seroprevalence of included studies.**

| NUMBER | 1. | 2. | 3. | 4. |
|---|---|---|---|---|
| REFERENCE | Appa A, Takahashi S, Rodriguez-Barraquer I, Chamie G, Sawyer A, Duarte E, et al. **Universal PCR and antibody testing demonstrate little to no transmission of SARS-CoV-2 in a rural community**. MedRxiv Prepr Serv Health Sci. 17 août 2020; | Angulo FJ, Finelli L, Swerdlow DL. **Estimation of US SARS-CoV-2 Infections, Symptomatic Infections, Hospitalizations, and Deaths Using Seroprevalence Surveys.** JAMA Netw Open. 4 janv 2021;4 (1):e2033706 | Arnold CRK, Srinivasan S, Herzog CM, Gontu A, Bharti N, Small M, et al. **SARS-CoV-2 Seroprevalence in a University Community: A Longitudinal Study of the Impact of Student Return to Campus on Infection Risk Among Community Members.** MedRxiv Prepr Serv Health Sci. 19 févr 2021; | Alessi D, Borré S, Barale A, Isabella A, Milano F, Rossi MA, et al. [**Seroprevalence of anti-SARS-CoV-2 IgG/IgM antibodies in Borgosesia (Piedmont Region, Northern Italy) population: a surveillance strategy in post-lockdown period?**]. Epidemiol Prev. déc 2020;44(5–6 Suppl 2):200-6. |
| LOCATION/ COUNTRY | Bolinas, California, the county of Marin | United States | Pennsylvanie | Borgosesia, (Piemonte region, northern Italy) |
| SELECTION CRITERIA | Men and women, 4 years old and over | Men and women, all ages | Men and women, 18 years old and over | Men and women, 18 years old and over, average age 55 Greater participation of women (54.4%) People with higher education (37.3%) People without specific previous symptoms (95.1%) |
| TYPE OF POPULATIONS | County residents and workers | General population | Residents living in a Centre at the University of Pennsylvania | General population |
| SAMPLE SIZE | 1,891 | April (n = 16,596), May (n = 14,291), June (n = 14,159), July (n = 12,367) August (n = 38,355) | 345 | 4,987 |
| SAMPLING METHOD | Random sampling | Random sampling | Random sampling | Random sampling |
| SEROPREVALENCE | 0.16% | 14.3% positive cases in the mid November | Of 345 community participants, 19 (5–5%) were positive for IgG SARS-CoV-2 antibodies. Of the 625 student participants who returned to campus for classes in the fall 195 (31–2%) were positive for SARS-CoV-2 antibodies. Twenty-eight (8–1%) of the community participants returned a positive IgG antibody result IgG antibody result by 9 December | 245 individuals tested positive for IgM or IgG, and the estimated prevalence was 4.9%. 209 out of 245 subjects who were positive on the rapid test underwent RT-PCR and this resulted in the isolation of 24 positive subjects |
| STUDY PERIOD | April-2020 | November-2020 | August-December 2020 | May-2020 |
| RISK FACTORS | | The most densely populated centres were the most affected by COVID-19 | | |
| SYMPTOMS | fever, cough, muscle aches, severe fatigue, difficulty breathing, diarrhea, loss of smell and/ or taste | | | Mild symptoms |
| NUMBER | 5. | 6. | 7. | 8. |
| REFERENCE | Aziz NA, Corman VM, Echterhoff AKC, Richter A, Schmandke A, Schmidt ML, et al. **Seroprevalence and correlates of SARS-CoV-2 neutralizing antibodies: Results from a population-based study in Bonn, Germany** [Internet]. 2020 Aug [cited 2022 Jan 17] p. 2020.08.24.20181206. Available from: https:// www.medrxiv.org/content/10.1101/2020.08.24. 20181206v1 | Babu GR, Sundaresan R, Athreya S, Akhtar J, Pandey PK, Maroor PS, et al. **The burden of active infection and anti-SARS-CoV-2 IgG antibodies in the general population: Results from a statewide survey in Karnataka, India** [Internet]. medRxiv; 2020 [cited 2022 Jan 26]. p. 2020.12.04.20243949. Available from: https://www.medrxiv.org/content/10.1101/2020.12.04. 20243949v1 | Bajema KL, Wiegand RE, Cuffe K, Patel SV, Iachan R, Lim T, et al. **Estimated SARS-CoV-2 Seroprevalence in the US as of September 2020**. JAMA Intern Med. 1 avr 2021;181(4):450-60. | Batista KBC, Caseiro MM, Barros CR, Martins LC, Chioro A, Araújo ESA de, et al. **COVID-19 Seroprevalence in Baixada Santista Metropolitan Area–Brazil** [Internet]. medRxiv; 2020 [cited 2022 Feb 2]. p. 2020.08.28.20184010. Available from: https://www.medrxiv.org/content/10.1101/2020.08.28. 20184010v1 |

*(Continued)*

**Table 1.** (Continued)

| NUMBER | 1. | 2. | 3. | 4. |
|---|---|---|---|---|
| LOCATION/ COUNTRY | City of Bonn, Germany | Karnataka, India | United States (50 states) | Baixada Santista Metropolitan Area, Brazil |
| SELECTION CRITERIA | People (women and men) aged 30 years and abovewho lived in Bonn, Germany The population was the population enrolled in the Rhineland study | Men and women, 18 years old and over | Men and women, all ages | Men and women, all ages |
| TYPE OF POPULATIONS | An ongoing community based prospective cohort study (adults in general population) | General population | General population | Probabilistic population-based sample |
| SAMPLE SIZE | 4,771 | 16,416 | 204,287 | 2,342 |
| SAMPLING METHOD | Based on geographical areas and volunteers Random sampling | Based on geographical areas (hospitals from districts) Random sampling | Random sampling | Randomized in each city and street selection based on data drawn from the 2010 Brazilian census Stratified by age, gender and living conditions |
| SEROPREVALENCE | 0.97% (95% CI: 0.72–1.30) by ELISA. 0.36% (95% CI: 0.21–0.61) by PRNT | IgG: 16·4% (95% CI: 15·1–17·7) | Jurisdictional seroprevalence over 4 collection periods collection periods ranged from less than 1% to 23%. In 42 of the 49 jurisdictions with sufficient samples to estimate to estimate seroprevalence in all periods, less than 10% of individuals had detectable SARS-CoV-2 antibodies | 1.4% at a 95%CI level (0.93–1.93) |
| STUDY PERIOD | 24. Avril-30. June 2020 | 3. September-16. September 2020 | September-2020 | February-2020 |
| RISK FACTORS | | Regions, age | | |
| SYMPTOMS | A recent history of reduced taste or smell, fever, chills/hot flashes, pain while breathing, pain in arms/legs, as well as muscle pain and weakness | | | |
| NUMBER | 9. | 10. | 11. | 12. |
| REFERENCE | Biggs HM, Harris JB, Breakwell L, Dahlgren FS, Abedi GR, Szablewski CM, et al. **Estimated Community Seroprevalence of SARS-CoV-2 Antibodies—Two Georgia Counties, April 28-May 3, 2020.** MMWR Morb Mortal Wkly Rep. 24 juill 2020;69(29):965-70. | Bobrovitz N., Krishan Arora R., Cao C., Boucher E., Liu M., Donnici C., et al. **"Global seroprevalence of SARS-CoV-2 antibodies: A systematic review and meta-analysis.** Plos One. 26 Dec 2021, 16(6):e0252617. | Bogogiannidou Z, Vontas A, Dadouli K, Kyritsi MA, Soteriades S, Nikoulis DJ, et al. **Repeated leftover serosurvey of SARS-CoV-2 IgG antibodies, Greece,** March and April 2020. Euro Surveill Bull Eur Sur Mal Transm Eur Commun Dis Bull. août 2020;25(31). | Borges LP, Martins AF, de Melo MS, de Oliveira MGB, Neto JM de R, Dósea MB, et al. **Seroprevalence of SARS-CoV-2 IgM and IgG antibodies in an asymptomatic population in Sergipe, Brazil.** Rev Panam Salud Publica Pan Am J Public Health. 2020;44:e108. |
| LOCATION/ COUNTRY | DeKalb and Fulton counties, in the metropolitan Atlanta area Atlanta, Georgia | Worldwide | Greece | The state of Sergipe, Brazil. |
| SELECTION CRITERIA | Men and women, all ages | Men and women, all ages | Men and women, 18 years old and over | Men (48.2%), Women (51.8%), all ages (average 39.76 years) |
| TYPE OF POPULATIONS | General population of these two counties | General population | General population | Asymptomatic general population |
| SAMPLE SIZE | 696 and 394 households | 9,329,185, in 968 serosurveys from 605 reports | 6,586 | 3,046 |
| SAMPLING METHOD | Two-stage clustering was used to randomly select 30 census blocks in each county | Several techniques | Geographical stratification, sampling on global units All of which gives a representative sample | Stratified sample |

(*Continued*)

**Table 1.** (Continued)

| NUMBER | 1. | 2. | 3. | 4. |
|---|---|---|---|---|
| SEROPREVALENCE | 19 positive cases of 696 i.e. 2.7% of the total | Between 0.6% in South East Asia and 59% Sub-Saharan Africa: 5.01% South Asia: 2.84% Central Europe, Eastern Europe, and Central Asia: 2.83% Latin America and Caribbean: 2.71% Southeast Asia, East Asia, and Oceania: 0.18% | 0.36% 24 positive cases out of 6.586 samples | 347 tested positive for IgM, 218 tested positive for IgG Women over 40 years of age had the highest prevalence for IgM |
| STUDY PERIOD | April-May 2020 | September 2019- December 2020 | March-April 2020 | May-2020 |
| RISK FACTORS | The most densely populated centres were the most affected by COVID-19 | Job type, patient's COVID 19 contact, deprived area, location, ethnicity/race, age | | |
| SYMPTOMS | Cough or shortness of breath, fever, loss of taste or smell | | Mild symptoms | Asymptomatic cases |
| NUMBER | 13. | 14. | 15. | 16. |
| REFERENCE | Carrat F, Lamballerie X de, Rahib D, Blanché H, Lapidus N, Artaud F, et al. **Seroprevalence of SARS-CoV-2 among adults in three regions of France following the lockdown and associated risk factors: a multicohort study** [Internet]. 2020 Sep [cited 2022 Jan 20] p. 2020.09.16.20195693. Available from: https://www.medrxiv.org/content/10.1101/2020.09.16.20195693v1 | Cerino P, Coppola A, Volzone P, Pizzolante A, Pierri B, Atripaldi L, et al. **Seroprevalence of SARS-CoV-2-specific antibodies in the town of Ariano Irpino (Avellino, Campania, Italy): a population-based study**. Future Sci Oa.:FSO673. | Chen X, Chen Z, Azman AS, Deng X, Sun R, Zhao Z, et al. **Serological evidence of human infection with SARS-CoV-2: a systematic review and meta-analysis.** Lancet Glob Health. 8 mars 2021; | Coyle PV, Chemaitelly H, Kacem MABH, Molawi NHAA, Kahlout RAE, Gilliani I, et al. **SARS-CoV-2 seroprevalence in the urban population of Qatar: An analysis of antibody testing on a sample of 112,941 individuals** [Internet]. 2021 Jan [cited 2022 Jan 20] p. 2021.01.05.21249247. Available from: https://www.medrxiv.org/content/10.1101/2021.01.05.21249247v1 |
| LOCATION/ COUNTRY | France | Ariano Irpino, Italy | European region, American region Western Pacific region | Qatar |
| SELECTION CRITERIA | Participants in a survey on COVID-19 from an existing consortium of three general adult population cohorts living in the Ile-de-France (IDF) or Grand Est (GE)—or in the Nouvelle-Aquitaine (NA) | Men and women, all ages | Men and women, 222 are between 18 and 69 years old, 7 under the age of 18 including 2 who are under 10 years old and one over 80 years old | Individuals receiving routine and other clinical care at Hamad Medical Corporation (HMC), a main provider of healthcare to the urban population of this country and the nationally designated provider for Coronavirus Disease 2019 (COVID-19) healthcare needs |
| TYPE OF POPULATIONS | Adults | Asymptomatic general population | General population, health care workers | Urban population |
| SAMPLE SIZE | 14,628 | 13,218 | 230 | 112,941 |
| SAMPLING METHOD | Identification of participants in a survey on COVID-19 from an existed consortium based on geographical areas Random sampling | Random sampling | Random sampling | Based on urban areas and medical, Care structures Random sampling |
| SEROPREVALENCE | Adjusted estimates (positive ELISA-S): 10.0% (95%CI 9.1%;10.9%) in IDF, 9.0% (95%CI 7.7%; 10.2%) in GE and 3.1% (95%CI 2.4%; 3.7%), in NA | A total of 738 citizens tested positive for anti-SRAS-CoV-2 antibodies (398 females, 340 males). The overall prevalence in the sample was 5.6%. Among the HIV-positive citizens, 101 cases were RT-PCR positive (0.76% of the total population). Among citizens aged 14–18 years, 18–65 years and >65 years, seroprevalence was equal to 6.1, 5.6 and 4%, respectively. In the paediatric cohort (<14 years), the seroprevalence was 13% | South East Asia (19.6%, 95% CI 5.5–33.6, all in India) African region (16.3%, 0.0–33.7%). Americas region (6.8%, 5.0–8.5) European Region (4.7%, 3.6–5.9). Western Pacific (1.7%) | 13.3% (95% CI: 13.1–13.6%) |

*(Continued)*

**Table 1.** (Continued)

| NUMBER | 1. | 2. | 3. | 4. |
|---|---|---|---|---|
| STUDY PERIOD | May-June 2020 | May-2020 | December 2019—September 2020 | 12. May-9. September 2020 |
| RISK FACTORS | Age and the number of child or adolescent lived in the same household, smoking status | | Job type, gender | Sex, age, nationality, clinical care type and testing date |
| SYMPTOMS | | Asymptomatic cases | | |
| NUMBER | 17. | 18. | 19. | 20 |
| REFERENCE | Della Valle P, Fabbri M, Madotto F, Ferrara P, Cozzolino P, Calabretto E, et al. **Occupational Exposure in the Lombardy Region (Italy) to SARS-CoV-2 Infection: Results from the MUSTANG-OCCUPATION-COVID-19 Study**. Int J Environ Res Public Health. mars 2021;18(5):2567. | Duan S, Zhou M, Zhang W, Shen J, Qi R, Qin X, et al. **Seroprevalence and asymptomatic carrier status of SARS-CoV-2 in Wuhan City and other places of China**. PLoS Negl Trop Dis. janv 2021;15(1): e0008975. | Doi A, Iwata K, Kuroda H, Hasuike T, Nasu S, Kanda A, et al. **Seroprevalence of novel coronavirus disease (COVID-19) in Kobe, Japan** [Internet]. medRxiv; 2020 [cited 2022 Jan 31]. p. 2020.04.26.20079822. Available from: https://www.medrxiv.org/content/10.1101/2020.04.26.20079822v2 | Feehan AK, Fort D, Garcia-Diaz J, Price-Haywood E, Velasco C, Sapp E, et al. **Point prevalence of SARS-CoV-2 and infection fatality rate in Orleans and Jefferson Parish, Louisiana, May 9–15, 2020** [Internet]. 2020 Jun [cited 2022 Jan 14] p. 2020.06.23.20138321. Available from: https://www.medrxiv.org/content/10.1101/2020.06.23.20138321v1 |
| LOCATION/ COUNTRY | Lombardy region, Italy | China (various rights) | Kobe, Japan | New Orleans, Louisiana, USA |
| SELECTION CRITERIA | Men and women, 18 years old and over | Men and women, all ages | Patients who visited outpatient clinics of the hospital and received blood testing for any reason and not visited the emergency department or the designated fever consultation service | Population representative of the demographics of the Parishes |
| TYPE OF POPULATIONS | Workers | Healthy general population, asymptomatic cases | General population | Population who lived in Orleans and Jefferson Parishes (representative pool of the demographics of the Parishes) |
| SAMPLE SIZE | 2,255 | 63,107 | 1,000 | 2,640 |
| SAMPLING METHOD | Random sampling | Random sampling | Random sampling | Based on geographical aeras, the use of a novel two-step system for representative sample and volunteers with stratification. |
| SEROPREVALENCE | 4.8% tested positive for IgG/IgM antibodies to SARS-CoV-2 antibody test, of which 81.7% were IgG positive only | The total positive SARS-CoV-2 IgG and IgM antibody level was 1.68% (186/11.086) in WH, 0.59% (226/38.171) in Hubei province without Wuhan (HB), 0.38% (53/13.850) in the country excluding Hubei province respectively. The positive IgM rate was 0.46% (51/11.086) in WH 0.13% (51/38.171) in HB and 0.07% (10/13.850) in CN. The incidence of positive IgM levels in healthy individuals increased from 6 March to 3. May 2020 in women and the elderly had a higher probability of being infected than men or younger people | 3.3%, 95% CI: 2.3–4.6% | Weight seroprevalence 6.86% (95% CI: 6.8–6.91) |
| STUDY PERIOD | May-October 2020 | March-May 2020 | 31. March-7. April 2020 | 9. May-15. May 2020 |
| RISK FACTORS | | Children | | Ethnicity |

(*Continued*)

**Table 1.** (Continued)

| NUMBER | 1. | 2. | 3. | 4. |
|---|---|---|---|---|
| SYMPTOMS | Cough, sore throat, headache, muscles, anosmia/ageusia, respiratory distress, chest pain, tachycardia, gastrointestinal disorders, conjunctivitis, clinical diagnosis of pneumonia | Asymptomatic cases | | |
| NUMBER | 21. | 22. | 23. | 24. |
| REFERENCE | Feehan AK, Velasco C, Fort D, Burton JH, Price-Haywood E, Katzmarzyk PT, et al. **Racial and workplace disparities in seroprevalence of SARS-CoV-2 in Baton Rouge, Louisiana, July 15–31, 2020** [Internet]. 2020 Sep [cited 2022 Jan 20] p. 2020.08.26.20180968. Available from: https://www.medrxiv.org/content/10.1101/2020.08.26.20180968v1 | Figar S, Pagotto V, Luna L, Salto J, Manslau MW, Mistchenko AS, et al. Community-level **SARS-CoV-2 Seroprevalence Survey in urban slum dwellers of Buenos Aires City, Argentina: a participatory research** [Internet]. medRxiv; 2020 [cited 2022 Jan 31]. p. 2020.07.14.20153858. Available from: https://www.medrxiv.org/content/10.1101/2020.07.14.20153858v2 | Franceschi VB, Santos AS, Glaeser AB, Paiz JC, Caldana GD, Lessa CLM, et al. **Population-based prevalence surveys during the COVID-19 pandemic: a systematic review** [Internet]. medRxiv; 2020 [cited 2022 Feb 9]. p. 2020.10.20.20216259. Available from: https://www.medrxiv.org/content/10.1101/2020.10.20.20216259v1 | Gégout-Petit A, Jeulin H, Legrand K, Jay N, Bochnakian A, Vallois P, et al. **"Seroprevalence of SARS-CoV-2, Symptom Profiles and Sero-Neutralization during the first COVID-19 wave in suburban area, France**. MedRxiv 2021.02.10.21250862; doi:https://doi.org/10.1101/2021.02.10.21250862 |
| LOCATION/ COUNTRY | Baton Rouge, Louisiana | Buenos Aires City, Argentina | Europe: 15 (40.5%) North America: 8 (21.6%) South America:8 (21.6%) Asia: 5 (13.5%) Africa: 1 (2.7%). In all, 19 countries | Nancy, France |
| SELECTION CRITERIA | Residents living in eligible ZIP codes and giving consent | Men and women, 14 years old and over | Men and women, all ages | Men and women, 5 years old and over |
| TYPE OF POPULATIONS | General population | Community (households, families) | General population | General population |
| SAMPLE SIZE | 2,138 | 873 | 37 surveys (394,090 + 9,899,828 with the Wuhan's screening program | 1,11 |
| SAMPLING METHOD | Randomized residents targeted with digital ads for recruitment, then selection based on volunteers and re-stratification by census designation | Selected from a probabilistic sample of households | Random sampling | Randomly sampled from electoral lists and invited with family members |
| SEROPREVALENCE | 3.6% (2.8,4.4) | 53.4% (95%IC 52.8% to 54.1%) | 19.6% to 69% | 2.1% |
| STUDY PERIOD | 15. July-31. July 2020 | 10. June- 1. July 2020 | 15. July-5. September 2020 | June 2020 |
| RISK FACTORS | Job type | Areas (deprived areas) | Ethnicity/Race, job type, location, deprived area | |
| SYMPTOMS | | | | Cough, fatigue, shortness of breath, aches anosmia/ageusia, muscle pain, sore throat, headaches, rhinorrhea, chest pain, diarrhea, abdominal pain, loss of balance, nausea, appetite loss, skin rashes, irritated eye |
| NUMBER | 25 | 26 | 27 | 28 |
| REFERENCE | George CE, Inbaraj LR, Chandrasingh S, de Witte LP. **High seroprevalence of COVID-19 infection in a large slum in South India; what does it tell us about managing a pandemic and beyond?** Epidemiol Infect. 4 févr 2021;149:e39. | Ghose A, Bhattacharya S, Karthikeyan AS, Kudale A, Monteiro JM, Joshi A, et al. **Community prevalence of antibodies to SARS-CoV-2 and correlates of protective immunity in an Indian metropolitan city** [Internet]. medRxiv; 2020 [cited 2022 Jan 31]. p. 2020.11.17.20228155. Available from: https://www.medrxiv.org/content/10.1101/2020.11.17.20228155v2 | Grant R, Dub T, Andrianou X, Nohynek H, Wilder-Smith A, Pezzotti P, et al. **SARS-CoV-2 population-based seroprevalence studies in Europe: a scoping review**. BMJ Open. 1 avr 2021;11(4):e045425. | Hallal PC, Hartwig FP, Horta BL, Silveira MF, Struchiner CJ, Vidaletti LP, et al. **SARS-CoV-2 antibody prevalence in Brazil: results from two successive nationwide serological household surveys**. Lancet Glob Health. nov 2020;8(11):e1390-8. |
| LOCATION/ COUNTRY | South India | India (a metropolitan city) | Europe | Brazil (133 cities) |

(Continued)

**Table 1.** (Continued)

| NUMBER | 1. | 2. | 3. | 4. |
|---|---|---|---|---|
| SELECTION CRITERIA | Men and women, 18 years and over, (74.3%) were women | Live in an inactive containment zone, healthy subjects who not presented fever, not had an illness episode including COVID-19 in the past | Men and women, all ages | Men and women, all ages |
| TYPE OF POPULATIONS | Population from a large slum with about one third of the population reported comorbidities. | Adults living in a metropolitan city | General population | General population |
| SAMPLE SIZE | 499 | 1,659 | 194 | 25,025 (14–21 May), 31,165 (4–7 June) |
| SAMPLING METHOD | Random sampling | Sub-wards randomly selected and Stratified by incidence (based on areas and incidence) | Random sampling | Random sampling (in the list of members of the households provided) |
| SEROPREVALENCE | The overall seroprevalence of IgG antibody to COVID-19 was 57.9% The ratio of undetected cases to infections was 1: 195 and the case fatality rate was calculated as 2.94 per 10.000 infections | (51·3%; 95%CI 39·9 to 62·4) | Between 0.42% and 23.3% positives cases Residual clinical: Greece: 0.42% Germany: 13.6% Blood Donors: Northwest Germany: 0.91% Lombardy, Italy: 23.3% | 9.259 positive cases of 25.025 i.e. 37% |
| STUDY PERIOD | October- 2020 | 20. July-5. August 2020 | January-September 2020 | May-June 2020 |
| RISK FACTORS | | Living conditions, age | The most densely populated centres were the most affected by COVID-19 | The most densely populated centres were the most affected by COVID-19 Age, Young people (between 18–30 years old) |
| SYMPTOMS | | | | |
| NUMBER | 29. | 30. | 31. | 32. |
| REFERENCE | Havers FP, Reed C, Lim T, Montgomery JM, Klena JD, Hall AJ, et al. **Seroprevalence of Antibodies to SARS-CoV-2 in 10 Sites in the United States,** March 23-May 12, 2020. JAMA Intern Med. 21 juill 2020; | He Z, Ren L, Yang J, Guo L, Feng L, Ma C, et al**. Seroprevalence and humoral immune durability of anti-SARS-CoV-2 antibodies in Wuhan, China: a longitudinal, population-level, cross-sectional study**. Lancet. 20 mars 2021;397(10279):1075-84. | Hozé N, Paireau J, Lapidus N, Tran Kiem C, Salje H, Severi G, et al. **Monitoring the proportion of the population infected by SARS-CoV-2 using age-stratified hospitalisation and serological data: a modelling study.** Lancet Public Health. 8 avr 2021; | Husby A, Corn G, Krause TG. **SARS-CoV-2 infection in households with and without young children: Nationwide cohort study** [Internet]. 2021 Mar [cited 2022 Jan 20] p. 2021.02.28.21250921. Available from: https://www.medrxiv.org/content/10.1101/2021.02.28.21250921v1 |
| LOCATION/ COUNTRY | United States | Wuhan, China | France | Denmark |
| SELECTION CRITERIA | Men and women, 18 years old or younger and over 65 years old | Men and women, all ages, asymptomatic cases | Men and women, 20 years old and over | Men and women, 18–60 years old |
| TYPE OF POPULATIONS | General population | General population | Proportion of adults infected with SARS-CoV-2 and the proportion of infections detected in the two most affected regions in France in May 2020, applied our approach to the 13 French metropolitan regions over the period | We constructed a cohort of SARS-CoV-2-test-positive individuals who were followed up for hospitalization until 30 days after positive test. |
| SAMPLE SIZE | 16,025 | 9,542 from 3,556 families | 9,782 | 449,915 living in households with young children, while among 2,629,821 without young children in their household |
| SAMPLING METHOD | A practical sample of residual seraglio | Random sampling Those who had been living in Wuhan for at least 14 days since 1 December 2019 | Random sampling | Random sampling |

(*Continued*)

**Table 1.** (Continued)

| NUMBER | 1. | 2. | 3. | 4. |
|---|---|---|---|---|
| SEROPREVALENCE | 1,0% to6,9% | Between 5·and 6% | 5–7% (95% CI 5-1-6-4) of adults in metropolitan France had been infected with SARS-CoV-2 by 11 May 2020. This proportion remained stable until August 2020 and increased to 14–9% (13-2-16-9) by 15 January 2021. With 26–5% (23-4-29-8) of adult residents having been infected in Île-de-France (Paris region) compared to 5–1% (4-5-5-8) in Brittany in January 2021, regional variations remained significant (coefficient of variation [CV] 0–50) although less so than in May 2020 (CV 0–74). The proportion infected was twice as high (20–4%, 15-6-26-3) among 20–49 year olds than among those aged 50 years or more (9–7%, 6-9-14-1). 40–2% (34-3-46-3) of adult infections were detected in June to August 2020, compared to 49–3% (42-9-55-9) in November, 2020, to January, 2021 | -Among 449.915 samples living in households with young children, 5.761 (1.28%)were tested positive for SARS-CoV-2<br>-Among 2.629.821 adults without young children in their household, 33.788 (1.28%) were tested positive for SARS-CoV-2 (adjusted hazard ratio, 1.05; 95% confidence interval, 1.02 to 1.09) |
| STUDY PERIOD | March-May 2020 | April-December 2020 | March 2020-February 2021 | 27. February-15. November 2020 |
| RISK FACTORS | | | | Age, gender, with or without children at household (the number), ethnicity, comorbidities (Asthma, Chronic pulmonary disease, Cardiovascular disease, Diabetes mellitus, Inflammatory bowel disease Malignancy, Renal failure) |
| SYMPTOMS | Mild symptoms | Respiratory symptoms, asymptomatic cases | | |
| NUMBER | 33. | 34. | 35. | 36. |
| REFERENCE | Hussein NR, Balatay AA, Naqid IA, Jamal SA, Rasheed NA, Ahmed AN, et al. **COVID-19 antibody seroprevalence in Duhok, Kurdistan Region, Iraq: A population-based study** [Internet]. medRxiv; 2021 [cited 2022 Feb 9]. p. 2021.03.23.21254169. Available from: https://www.medrxiv.org/content/10.1101/2021.03.23.21254169v1 | Inbaraj LR, George CE, Chandrasingh S. **Seroprevalence of COVID-19 infection in a rural district of South India: A population-based seroepidemiological study**. PloS One. 2021;16(3):e0249247. | Investigators A to beat coronavirus/Action pour battre le coronavirus (Ab-CS, Jha P**. COVID Seroprevalence, Symptoms and Mortality During the First Wave of SARS-CoV-2 in Canada** [Internet]. 2021 Mar [cited 2022 Jan 20] p. 2021.03.04.21252540. Available from: https://www.medrxiv.org/content/10.1101/2021.03.04.21252540v1 | Investigators RS, Riley S, Ainslie KEC, Eales O, Jeffrey B, Walters CE, et al. **Community prevalence of SARS-CoV-2 virus in England during May 2020: REACT study** [Internet]. medRxiv; 2020 [cited 2022 Feb 2]. p. 2020.07.10.20150524. Available from: https://www.medrxiv.org/content/10.1101/2020.07.10.20150524v1 |
| LOCATION/ COUNTRY | Duhok City, Iraq | South India | Canada | United Kingdom, England |
| SELECTION CRITERIA | Men and women, all ages | Men and women, 18 years old and over | Men and women, 18 years old and over | Men and women, 5 years old and over |
| TYPE OF POPULATIONS | We included samples from both urban and rural populations (General population)<br>A person was eligible to be included in the study if the person was at least 16 years old, was a resident of one of the considered districts, had been living at a specific address for at least 6 months | Population of two grampanchayats (administrative group of 5 to 8 villages). Four sub-districts | General population, national demographic Distribution and Forum panel members who lived in 17 pre-defined public health regions | General population: Participants were randomly selected from the National Health Service (NHS) list of patients registered with a general practitioner |

(*Continued*)

**Table 1.** (Continued)

| NUMBER | 1. | 2. | 3. | 4. |
|---|---|---|---|---|
| SAMPLE SIZE | 743 | 509 | We surveyed a representative sample of 19,994 Canadians about COVID symptoms and analyzed IgG antibodies against SARS-CoV-2 from self-collected dried blood spots in 8,967 samples | 120,610 |
| SAMPLING METHOD | Random sampling | Random sampling | Stratified samples | Random sampling |
| SEROPREVALENCE | Among the participants, 465/743 (62.58%) tested positive for antibodies. Among the participants with antibodies, 262/465 (56.34%) denied having any history of COVID-19 related symptoms | The age and gender adjusted seroprevalence was 8.5%. The unadjusted seroprevalence in participants with hypertension and diabetes was 16.3% and 10.7% respectively<br>When we adjusted for test performance, the seroprevalence was 6.1%<br>The study estimated that 7 undetected infected persons for each case confirmed by RT-PCR<br>The infection mortality rate (IFR) was calculated to be 12.38 per 10.000 infections as of 22 October 2020 | The best estimate of adult seroprevalence nationally is 1.7%, but as high as 3.5% depending on assay cut-offs. The highest prevalence was in Ontario (2.4–3.9%) and in younger adults aged 18–39 years (2.5–4.4%). | We found 159 positives from 120.610 swabs giving an average prevalence of 0.13% (95% CI: 0.11%,0.15%)<br>Adults aged 18 to 24 yrs had the highest swab-positivity rates, while those >64 yrs had the lowest |
| STUDY PERIOD | July 2020-February 2021 | May-September 2020 | May-September 2020 | 1. May-1. June 2020 |
| RISK FACTORS | Gender, marital status, chronic disease | | Gender, age, location, education, household size, ethnicity, smocking | Age, ethnicity, location, job |
| SYMPTOMS | Fever, myalgia, loss of smell and taste, shortness of breath, cough, diarrhea, headache)<br>A total of 203/465 (43.66%) participants had a history of different symptoms | | Fever, difficult breathing, dry cough, loss of smell, "Covid toe" | Asymptomatic cases, nausea and/or vomiting, diarrhoea, blocked nose, loss of smell, loss of taste, headache, chills and severe fatigue |
| NUMBER | 37. | 38. | 39. | 40. |
| REFERENCE | Javed W, Baqar JB, Abidi SHB, Farooq W**. Seroprevalence Findings from Metropoles in Pakistan: Implications for Assessing COVID-19 Prevalence and Case-fatality within a Dense**, Urban Working Population [Internet]. medRxiv; 2020 [cited 2022 Feb 9]. p. 2020.08.13.20173914. Available from: https://www.medrxiv.org/content/10.1101/2020.08.13.20173914v1 | Jõgi P, Soeorg H, Ingerainen D, Soots M, Lättekivi F, Naaber P, et al. **Seroprevalence of SARS-CoV-2 IgG antibodies in two regions of Estonia (KoroSero-EST-1)** [Internet]. 2020 Oct [cited 2022 Jan 20] p. 2020.10.21.20216820. Available from: https://www.medrxiv.org/content/10.1101/2020.10.21.20216820v1 | Khalagi K, Gharibzadeh S, Khalili D, Mansournia MA, Samiee SM, Aghamohamadi S, et al. **Prevalence of COVID-19 in Iran: Results of the first survey of the Iranian COVID-19 Serological Surveillance program** [Internet]. medRxiv; 2021 [cited 2022 Feb 9]. p. 2021.03.12.21253442. Available from: https://www.medrxiv.org/content/10.1101/2021.03.12.21253442v2 | Kshatri JS, Bhattacharya D, Kanungo S, Giri S, Palo SK, Parai D, et al. **Findings from serological surveys (in August 2020) to assess the exposure of adult population to SARS Cov-2 infection in three cities of Odisha, India** [Internet]. medRxiv; 2020 [cited 2022 Feb 9]. p. 2020.10.11.20210807. Available from: https://www.medrxiv.org/content/10.1101/2020.10.11.20210807v1 |
| LOCATION/COUNTRY | Metropoles of Pakistan (Karachi, Lahore, Multan, Peshawar and Quetta) | In two regions (Harju county and in Saaremaa), Estonia | Iran (all provinces) | Odisha, Eastern India |
| SELECTION CRITERIA | Men and women, 18–65 years old | Men and women, 0–100 years old | Men and women, 6 years old and over | Men and women, 18 years old and over |
| TYPE OF POPULATIONS | General population<br>The sample size included an adult, working population aged 18–65 years, recruited from dense, urban workplaces including factories, corporates, restaurants, media houses, schools, banks, healthcare providers in hospitals, and families of positive cases | Patients of two general practitioner in Estonia. | General population | General population, residing in the city since at least the past 3 months. |
| SAMPLE SIZE | 24,210 | 1,960 | 11,256 | 4,146 |
| SAMPLING METHOD | Random sampling | Stratified random sampling | Stratified random sampling | Multistage random sampling |

(*Continued*)

**Table 1.** (Continued)

| NUMBER | 1. | 2. | 3. | 4. |
|---|---|---|---|---|
| SEROPREVALENCE | The study results reveal that from 24.210 individuals screened, 17.5% tested positive, with 7% IgM positive, 6.0% IgG positive and 4.5% combined IgM and IgG positive. | Seroprevalence was 1.5% (95% confidence interval (CI) 0.9–2.5) and 6.3% (95% CI 5.0–7.9), infection fatality rate 0.1% (95% CI 0.0–0.2) and 1.3% (95% CI 0.4–2.1) in Tallinn and Saaremaa, respectively. | Prevalence of COVID-19 until August 20, 2020 was estimated as 14.2% (95% uncertainty interval: 13.3%, 15.2%), which was equal to 11.958.346 (95% confidence interval: 11.211.011–12.746.776) individuals. The prevalence of infection was 14.6%, 13.8%, 16.6%, 11.7%, and 19.4% among men, women, urban population, rural population, and individuals ≥60 years of age, respectively | Berhampur: 31.14% (95% CI:28.69–33.66%) Rourkela: 24.59% (95% CI:22.39–26.88%) Bhubaneswar: 5.24% (95% CI:4.10–6.58%) |
| STUDY PERIOD | July-2020 | 8. May-31. July 2020 | August-October 2020 | August-2020 |
| RISK FACTORS | Job, age, location, family contact cases | Age, gender, household size, contact with confirmed case Of seropositive subjects 19.2% (14/73) had acute respiratory illness. IgG concentrations were higher if fever, difficulty breathing, shortness of breath, chest pain or diarrhea was present, or hospitalization required | Gender, age, location (urban/rural population) | Gender (females reported a higher seroprevalence with 22.8% as compared to males with 18.8%) |
| SYMPTOMS | Mild symptoms | Fever, diarrhea and the absence of cough and runny nose were associated with seropositivity in individuals aged 50 or more years | Mild symptoms, asymptomatic cases | Majority Asymptomatic with 93.87% Symptoms: -Fever (68.89%) -Cough (46.06%) -Myalgia (32.67%) |
| NUMBER | 41. | 42. | 43 | 44. |
| REFERENCE | Ladage D, Höglinger Y, Ladage D, Adler C, Yalcin I, Braun RJ. **SARS-CoV-2 antibody prevalence and symptoms in a local Austrian population** [Internet]. 2020 Nov [cited 2022 Jan 20] p. 2020.11.03.20219121. Available from: https://www.medrxiv.org/content/10.1101/2020.11.03.20219121v1 | Lastrucci V, Lorini C, Del Riccio M, Gori E, Chiesi F, Sartor G, et al. **SARS-CoV-2 Seroprevalence Survey in People Involved in Different Essential Activities during the General Lock-Down Phase in the Province of Prato (Tuscany, Italy).** Vaccines. 19 déc 2020;8(4). | Lavezzo E, Franchin E, Ciavarella C, Cuomo-Dannenburg G, Barzon L, Del Vecchio C, et al. **Suppression of a SARS-CoV-2 outbreak in the Italian municipality of Vo'.** Nature. 2020 Aug;584(7821):425–9. | Levorson RE, Christian E, Hunter B, Sayal J, Sun J, Bruce SA, et al. **SARS-CoV-2 Seroepidemiology in Children and Adolescents** [Internet]. medRxiv; 2021 [cited 2022 Jan 31]. p. 2021.01.28.21250466. Available from: https://www.medrxiv.org/content/10.1101/2021.01.28.21250466v1 |
| LOCATION/ COUNTRY | Austria | Province of Prato, (Tuscany, Italy) | Italia | Northern Virginia, United States |
| SELECTION CRITERIA | Inhabitants of an Austrian township with a reported higher incidence for COVID-19 infection | Men and women, all ages | Men and women, all ages | Children and adolescents ≤19 years |
| TYPE OF POPULATIONS | General population (inhabitants of a township) | People who had carried out essential activities during the during the period of confinement | Population residents (General population) | Children and adolescents and Residence in Virginia |
| SAMPLE SIZE | 835 | 4,656 | 5,155 | 1,038 |
| SAMPLING METHOD | Based on location (towns) and incidence, then voluntary (recruiting with a Public call supported by local authorities) | Random sampling | Random sampling | Recruited from three settings in Northern Virginia (non-emergency health care settings, self-referral, the Inova Children's Hospital Pediatric Emergency Department) |

(*Continued*)

**Table 1.** (Continued)

| NUMBER | 1. | 2. | 3. | 4. |
|---|---|---|---|---|
| SEROPREVALENCE | 9% | 138positive cases<br>i.e. 2.96% | A total of 73 out of the 2.812 participants who were tested at the first survey were positive, which gives a prevalence of 2.6% (95% CI: 2.1–3.3%). The second survey identified 29 total positive cases (prevalence of 1.2%; 95% CI: 0.8–1.8%), 8 of which were new cases (prevalence of 0.3%; 95% CI: 0.15–0.7%) | 8.5% |
| STUDY PERIOD | June-2020 | May-2020 | February-March 2020 | 31. July-13. October 2020 |
| RISK FACTORS | | | Age, sex | Ethnicity, public or absent insurance, a history of COVID-19 symptoms, exposure to person with COVID-19, a household member positive for SARS-CoV-2 and multi-family or apartment dwelling without a private entrance. |
| SYMPTOMS | | | Asymptomatic cases, fever, cough, headache, sorethroat, diarrhea, malaise, conjunctivitis | A history of COVID-19 symptoms |
| NUMBER | 45. | 46. | 47. | 48. |
| REFERENCE | Levesque J, Maybury DW. **A note on COVID-19 seroprevalence studies: a meta-analysis using hierarchical modelling** [Internet]. 2020 May [cited 2022 Jan 14] p. 2020.05.03.20089201. Available from: https://www.medrxiv.org/content/10.1101/2020.05.03.20089201v1 | Lim T, Delorey M, Bestul N, Johannsen M, Reed C, Hall AJ, et al. **Changes in SARS CoV-2 Seroprevalence Over Time in Ten Sites in the United States, March—August, 2020**. Clin Infect Dis Off Publ Infect Dis Soc Am. 26 févr 2021; | Ling R, Yu Y, He J, Zhang J, Xu S, Sun R, et al. **Seroprevalence and epidemiological characteristics of immunoglobulin M and G antibodies against SARS-CoV-2 in asymptomatic people in Wuhan, China: a cross-sectional study** [Internet]. 2020 Aug [cited 2022 Jan 20] p. 2020.06.16.20132423. Available from: https://www.medrxiv.org/content/10.1101/2020.06.16.20132423v3 | Malani A, Shah D, Kang G, Lobo GN, Shastri J, Mohanan M, et al. **Seroprevalence of SARS-CoV-2 in slums and non-slums of Mumbai, India, during June 29-July 19, 2020** [Internet]. 2020 Sep [cited 2022 Jan 14] p. 2020.08.27.20182741. Available from: https://www.medrxiv.org/content/10.1101/2020.08.27.20182741v1 |
| LOCATION/ COUNTRY | European (Gangelt, Germani/Geneva, Switzerland) American locations (Chelsea, Massachusetts/ San Miguel County, Colorado/ LosAngeles County, California) | United States (10 sites) | Wuhan, China | Mumbai, India |
| SELECTION CRITERIA | Men and women, all ages | Men and women, all ages | All ages with no fever, headache or other symptoms of COVID-19 and residents in Wuhan | Men and women, 12 years old and over |
| TYPE OF POPULATIONS | General population | General population | Asymptomatic general population | People from slum and non-slum communities |
| SAMPLE SIZE | 9,663 | 1,800 | 18,712 | 6,609 (4,202 from slums and 2,702 from non-slums) |
| SAMPLING METHOD | Random sampling | A convenient sample of residual seraglio,<br>Random sampling | Open screening proposed to all residents of Wuhan, screening carried out in hospital | Random sampling |
| SEROPREVALENCE | Geneva: In the week of April 6 to 10, researchers found that 3.5% of the 343 participants tested positive, which we interpret as 12 individuals. In the week of April 14 to 17, they found that 5.5% of the 417 participants tested positive, which we interpret as 23 individuals<br>Chelsea:63 tested positive of 200 samples (31,5%)<br>San Miguel County: 96 total positive tests, out of 4,757 (2.01%)<br>Santa Clara County: 2.81%<br>Los Angeles County: Out of 863 participants, 4.1% tested positive, which we interpret as 35 individuals. The researchers reported the 95% confidence bounds on the prevalence of 2.8%, and 5.6% | Seroprevalence remained below 10% in all sites except New York and Florida, where it reached 23.2% and 13.3% respectively. New York and Florida, where it reached 23.2% and 13.3% respectively. | Age and gender adjust: 3.27(3.02–3.52)<br>Assay adjust: 2.72 (2.49–2.95) | The positive test rate was 54.1% (95% CI: 52.7 to 55.6) and 16.1% (95% CI: 14.9 to 17.4) in slums and non-slums, respectively, a difference of 38 percentage points (P < 0.001). Accounting for imperfect accuracy of tests (e.g., sensitivity, 0.90; specificity 1.00), seroprevalence was as high as 58.4% (95% CI: 56.8 to 59.9) and 17.3% (95% CI: 16 to 18.7) in slums and non-slums, respectively |

(*Continued*)

**Table 1.** (Continued)

| NUMBER | 1. | 2. | 3. | 4. |
|---|---|---|---|---|
| STUDY PERIOD | 30. March-21. April 2020 | March-August 2020 | 25. March- 28. April 2020 | 29. June-19. July 2020 |
| RISK FACTORS | | | Sex, geographic areas, different types of workplaces | Location, age, gender |
| SYMPTOMS | | | Asymptomatic cases | |
| NUMBER | 49. | 50. | 51. | 52. |
| REFERENCE | McLaughlin CC, Doll MK, Morrison KT, McLaughlin WL, O'Connor T, Sholukh AM, et al. **High Community SARS-CoV-2 Antibody Seroprevalence in a Ski Resort Community, Blaine County, Idaho, US. Preliminary Results**. MedRxiv Prepr Serv Health Sci. 21 juill 2020; | Melotti R, Scaggiante F, Falciani M, Weichenberger CX, Foco L, Lombardo S, et al**. Prevalence and determinants of serum antibodies to SARS-CoV-2 in the general population of the Gardena Valley** [Internet]. 2021 Mar [cited 2022 Jan 20] p. 2021.03.19.21253883. Available from: https://www.medrxiv.org/content/10.1101/2021.03.19.21253883v1 | Menachemi N, Yiannoutsos CT, Dixon BE, Duszynski TJ, Fadel WF, Wools-Kaloustian KK, et al. **Population Point Prevalence of SARS-CoV-2 Infection Based on a Statewide Random Sample—Indiana,** April 25–29, 2020. MMWR Morb Mortal Wkly Rep. 24 juill 2020;69(29):960-4. | Merkely B, Szabó AJ, Kosztin A, Berényi E, Sebestyén A, Lengyel C, et al. **Novel coronavirus epidemic in the Hungarian population, a cross-sectional nationwide survey to support the exit policy in Hungary**. GeroScience 2020;42:1063–74. |
| LOCATION/ COUNTRY | Blaine County, Idaho, United States | Gardena Valley | Indiana | Hungary |
| SELECTION CRITERIA | No criteria, all the population of the county concerned, 18 years and older | Selected with known extraction probability from the municipality registries, excluding nursing homes | Elderly population and in racial and ethnic minority communities | Men and women, 14 years old and over |
| TYPE OF POPULATIONS | The residents of this county | General population | Elderly samples and in racial and ethnic minority communities | Private households |
| SAMPLE SIZE | 917 | 2,244 | 3,658 | 10,474 |
| SAMPLING METHOD | Selection from volunteers who have registered via a secure web link, using the weighting of the pre-stratification to the distribution of the population by age and gender in each postal code | One-stage random sampling design stratified by municipality, gender and age group | Random sampling | Two-stage stratified probability sample of individuals was selected from the population registry, |
| SEROPREVALENCE | The range of seroprevalence after correction for potential selection bias was 21.9% to 24.2% That is 208 out of 917 is positive | 26.9% (95% confidence interval: 25.2%, 28.6%) in June 2020 | 1,74 | 3 had positive PCR and 69 had positive serological test |
| STUDY PERIOD | April-June 2020 | 26. May-8. June 2020 | April-2020 | March-2020 |
| RISK FACTORS | | Place of residence, economic activity | | Sex, activity professional, Place of work during the epidemic, smocking, comorbidities (hypertension, heart disease, diabetes milletus, chronic pulmonary disease, chronic renal disease, chronic liver disease, curretn malignancy, immunodeficiency) |
| SYMPTOMS | | Loss of taste or smell, fever, difficulty in breathing, pain in the limbs, and weakness | | Fever, fatigue, body aches, cough, headache, sore throat, shortness of breath, abdominal pain, nausea/vomiting, diarrhea, loss smell or toaste |
| NUMBER | 53. | 54. | 55 | 56. |

(*Continued*)

**Table 1.** (Continued)

| NUMBER | 1. | 2. | 3. | 4. |
|---|---|---|---|---|
| REFERENCE | Murhekar MV, Bhatnagar T, Selvaraju S, Saravanakumar V, Thangaraj JWV, Shah N, et al. **SARS-CoV-2 antibody seroprevalence in India, August-September, 2020: findings from the second nationwide household serosurvey**. Lancet Glob Health. mars 2021;9(3):e257-66. | Murhekar MV, Bhatnagar T, Selvaraju S, Rade K, Saravanakumar V, Vivian Thangaraj JW, et al. **Prevalence of SARS-CoV-2 infection in India: Findings from the national serosurvey, May-June 2020**. Indian J Med Res. août 2020;152(1 & 2):48-60. | Naranbhai V, Chang CC, Beltran WFG, Miller TE, Astudillo MG, Villalba JA, et al. **High Seroprevalence of Anti-SARS-CoV-2 Antibodies in Chelsea, Massachusetts**. J Infect Dis. 13 nov 2020;222(12):1955-9. | Naushin S, Sardana V, Ujjainiya R, Bhatheja N, Kutum R, Bhaskar AK, et al. **Insights from a Pan India Sero-Epidemiological survey (Phenome-India Cohort) for SARS-CoV-2** [Internet]. 2021 Feb [cited 2022 Jan 14] p. 2021.01.12.21249713. Available from: https://www.medrxiv.org/content/10.1101/2021.01.12.21249713v2 |
| LOCATION/COUNTRY | India | India | Chelsea, Massachusetts | India (17 States and two Union Territories) |
| SELECTION CRITERIA | Men and women, 10 years old and older | Men and women, all ages | Men and women, all ages | Men and women, all ages |
| TYPE OF POPULATIONS | General population, in the same 700 villages or neighbourhoods in 70 districts in India that were included in the first serosurvey | General population | Asymptomatic general population | Adult subjects working in CSIR laboratories and their family members (General population) |
| SAMPLE SIZE | 29,082 | 30,283 households were visited, 28,000 samples were registered | 200 | 10,427 |
| SAMPLING METHOD | Random sampling | Random sampling | Anonymised sampling for convenience | Random sampling |
| SEROPREVALENCE | The adjusted weighted seroprevalence of SARS-CoV-2 IgG antibodies in individuals aged 10 years or older was 6–6% (95% CI 5-8-7-4). Among 15,084 randomly selected adults (one per household), the adjusted weighted seroprevalence was 7–1% (6-2-8-2) | The population-weighted seroprevalence after adjustment for test performance was 0.73% [95% confidence interval (CI): 0.34–1.13]. [95% confidence interval (CI): 0.34–1.13] | 31.5%, 24.8% (25/101) were positive and 60% were IgM+IgG-. | 10.14% |
| STUDY PERIOD | August-September 2020 | May-June 2020 | March-April 2020 | August-September 2020 |
| RISK FACTORS | | | The most densely populated centres were the most affected by COVID-19 Low income, presence of an seropositiv case in the household | Age, gender |
| SYMPTOMS | Mild symptoms | 486 cases (1.7%) respiratory symptoms | 50.5% reported no symptoms (asymptomatic cases) in the previous four weeks Symptoms in the Last 4 Weeks: The most common symptoms reported were: -Cough (26.5%), -Rhinitis (24.0%), -Sore throat (23.5%), -Myalgia (23.0%)). 13.0% reported a reduced sense of smell or taste. 16.9% thought they had or have had COVID-19 illness. | Asymptomatic Symtoms: -Fever (50%) -Loss of smell and taste (25%) |
| NUMBER | 57. | 58. | 59. | 60. |
| REFERENCE | Nawa N, Kuramochi J, Sonoda S, Yamaoka Y, Nukui Y, Miyazaki Y, et al. **Seroprevalence of SARS-CoV-2 IgG Antibodies in Utsunomiya City, Greater Tokyo, after first pandemic in 2020 (U-CORONA): a household- and population-based study** [Internet]. 2020 Jul [cited 2022 Jan 20] p. 2020.07.20.20155945. Available from: https://www.medrxiv.org/content/10.1101/2020.07.20.20155945v1 | Noh J, Danuser G. **Estimation of the fraction of COVID-19 infected people in US states and countries worldwide**. Plos One. 8 févr 2021;16(2):e0246772. | Pagani G, Conti F, Giacomelli A, Bernacchia D, Rondanin R, Prina A, et al. **Seroprevalence of SARS-CoV-2 significantly varies with age: Preliminary results from a mass population screening**. J Infect. déc 2020;81 (6):e10-2. | Pan Y, Li X, Yang G, Fan J, Tang Y, Hong X, et al. **Seroprevalence of SARS-CoV-2 immunoglobulin antibodies in Wuhan, China: part of the city-wide massive testing campaign**. Clin Microbiol Infect. févr 2021;27(2):253-7. |

(*Continued*)

**Table 1.** (Continued)

| NUMBER | 1. | 2. | 3. | 4. |
|---|---|---|---|---|
| LOCATION/ COUNTRY | Utsunomiya City, Tokio | United States | Castiglione d'Adda, south-east Milan, Italy | Wuhan, China |
| SELECTION CRITERIA | Men and women, all ages | Men and women, all ages | Men and women, all ages | Men and women, all age |
| TYPE OF POPULATIONS | General population (resident in Utsunomiya City) | General population | General population | General population |
| SAMPLE SIZE | 742 | | 509 | 61,437 |
| SAMPLING METHOD | Random sampling | Random sampling | Random sampling | Random cluster sampling |
| SEROPREVALENCE | 0.43% to 1.23% | Between 2,5 and-22,4% positives cases | 22.60% | A total of 1.470 individuals tested positive for at least one antiviral antibody. Of the positive individuals, 324 (0.53%, 95% CI 0.47–0.59) and 1.200 (1.95%, 95% CI 1.85–2.07) were positive for IgM and IgG immunoglobulin, respectively, and 54 (0.08%, 95% CI 0.07–0.12) were positive for both antibodies. The positive rate of female antibody carriers was higher than that of male counterparts (male to female ratio). Male counterparts (male-to-female ratio 0.75), particularly among older citizens (ratio of 0.18 in the 90+ age subgroup) This indicates a gender gap in seroprevalence. In addition, viral nucleic acid detection using real-time pcr showed 8 (0.013%, 95% CI 0.006–0.026) asymptomatic virus carriers |
| STUDY PERIOD | 14. June-5. July 2020 | March-April 2020 | May-June 2020 | May-2020 |
| RISK FACTORS | Age, gender, number cohabitant, residential district | | | |
| SYMPTOMS | Symptoms: -Afebrile | | Fever, cough, anosmia, dysgeusia, dispnea | |
| NUMBER | 61. | 62. | 63. | 64. |
| REFERENCE | Parker DM, Bruckner T, Vieira VM, Medina C, Minin VN, Felgner PL, et al. **Epidemiology of the early COVID-19 epidemic in Orange County, California: comparison of predictors of test positivity, mortality, and seropositivity** [Internet]. 2021 Jan [cited 2022 Jan 14] p. 2021.01.13.21249507. Available from: https://www.medrxiv.org/content/10.1101/2021.01.13.21249507v2 | Pérez-Olmeda M, Saugar JM, Fernández-García A, Pérez-Gómez B, Pollán M, Avellón A, et al. **Evolution of antibodies against SARS-CoV-2 over seven months: experience of the Nationwide Seroprevalence ENE-COVID Study in Spain** [Internet]. 2021 Mar [cited 2022 Jan 20] p. 2021.03.11.21253142. Available from: https://www.medrxiv.org/content/10.1101/2021.03.11.21253142v1 | Petersen MS, Strøm M, Christiansen DH, Fjallsbak JP, Eliasen EH, Johansen M, et al. **Seroprevalence of SARS-CoV-2-Specific Antibodies, Faroe Islands**. Emerg Infect Dis. nov 2020;26(11):2761-3. | Poljak M, Valenčak AO, Štrumbelj E, Vodičar PM, Vehovar V, Rus KR, et al. **Seroprevalence of SARS-CoV-2 in Slovenia: results of two rounds of a nationwide population study on a probability-based sample, challenges and lessons learned.** Clin Microbiol Infect Off Publ Eur Soc Clin Microbiol Infect Dis. 7 avr 2021; |
| LOCATION/ COUNTRY | Orange County, California | Spain | Faroe Islands | Slovenia |
| SELECTION CRITERIA | Men and women, all ages | Men and women, all ages | Men and women, 18 years old and over | Men and women, all ages |
| TYPE OF POPULATIONS | General population | The community-dwelling population in general population | General population | General population |

(*Continued*)

**Table 1.** (Continued)

| NUMBER | 1. | 2. | 3. | 4. |
|---|---|---|---|---|
| SAMPLE SIZE | 318,492 | 10,153 | 1,075 | 3,000 |
| SAMPLING METHOD | Random sampling | A two-stage sampling stratified | From the Faroe Islands population register, they sampled 1.500 people and invited them by letter to a clinic visit. Those unable to attend a test site received a home visit | Random sampling |
| SEROPREVALENCE | 36.816 tested positive for COVID-19 | 2.595 participants 44 (35.1% of participants) were positive for anti-nucleocapsid IgG in at least one round. In fourth round, anti-nucleocapsid and anti-RBD IgG were detected in 5.5% and 5.4% participants of the randomly selected sub-cohort, and in 26.6% and 25.9% participants with at least one previous positive result, respectively | 0,7% 7 positive cases of 1.075 samples | SARS-CoV-2 seroprevalence in Slovenia increased fourfold from late April to October/November 2020, mainly due to a devastating second wave. Between 2.78 and-4.29% |
| STUDY PERIOD | January 2020–16. August 2020 | 15. March-06. December 2020 | April-May 2020 | April-November 2020 |
| RISK FACTORS | Age, gender, race/ethnicity, % college degree, % with insurance | | | |
| SYMPTOMS | | | | |
| NUMBER | 65. | 66. | 67 | 68 |
| REFERENCE | Pollán M, Pérez-Gómez B, Pastor-Barriuso R, Oteo J, Hernán MA, Pérez-Olmeda M, et al. **Prevalence of SARS-CoV-2 in Spain (ENE-COVID): a nationwide, population-based seroepidemiological study.** Lancet Lond Engl. 22 août 2020;396(10250):535-44. | Polvere I, Parrella A, Casamassa G, D'Andrea S, Tizzano A, Cardinale G, et al. **Seroprevalence of Anti-SARS-CoV-2 IgG and IgM among Adults over 65 Years Old in the South of Italy.** Diagnostics. mars 2021;11(3):483. | Poustchi H, Darvishian M, Mohammadi Z, Shayanrad A, Delavari A, Bahadorimonfared A, et al. **SARS-CoV-2 antibody seroprevalence in the general population and high-risk occupational groups across 18 cities in Iran: a population-based cross-sectional study.** Lancet Infect Dis. avr 2021;21(4):473-81. | Pritsch M, Radon K, Bakuli A, Le Gleut R, Olbrich L, Guggenbüehl Noller JM, et al. **Prevalence and Risk Factors of Infection in the Representative COVID-19 Cohort Munich.** Int J Environ Res Public Health. 30 mars 2021;18(7). |
| LOCATION/ COUNTRY | Spain | Southern Italy | Iran | Munich, German |
| SELECTION CRITERIA | Men and women, all ages | Men and women, 65 years old and over | Men and women, 18 years old and over | Men and women, all ages |
| TYPE OF POPULATIONS | All residents were invited to participate, with asymptomatic cases | General population | General population | General population |
| SAMPLE SIZE | 61,075 | 1,383 | 8,902 | 5,313 |
| SAMPLING METHOD | Random sampling in two stages stratified by province size and the municipality | Random sampling | Random sampling | Representative sample, Random sampling |
| SEROPREVALENCE | The seroprevalence was 5–0% (95% CI 4-7-5-4) by the point-of-care test and 4–6% (4-3-5-0) byimmunoassay, with a specificity-sensitivity range of 3–7% (3-3-4-0; both tests positive) to 6–2% (5-8-6-6; either test positive), with no gender differences and lower seroprevalence in children under 10 years of age (<3–1% by point-of-care test) | The overall seroprevalence of anti-SRAS-CoV-2 antibodies was 4.70%. statistically significant differences between the sexes. Of these, 69.69% were IgM positive, 23.08% IgG and 9.23% were positive for both. | 17.10% | SARS-CoV-2 specific antibody seropositivity was 1.82% (95% confidence interval (CI) 1.28–2.37%). compared to 0.46% of officially registered pcr positive cases in Munich |
| STUDY PERIOD | April-May 2020 | September-2020 | April-June 2020 | April-June 2020 |
| RISK FACTORS | | | Disadvantaged areas | |

(*Continued*)

**Table 1.** (Continued)

| NUMBER | 1. | 2. | 3. | 4. |
|---|---|---|---|---|
| SYMPTOMS | Anosmia or three or more symptoms compatible with COVID-19 was 49·1%, asymptomatic cases | | -Anosmia 75·0%, <br>-Fever 60·8% | |
| NUMBER | 69. | 70. | 71. | 72. |
| REFERENCE | Qin X, Shen J, Dai E, Li H, Tang G, Zhang L, et al. **The seroprevalence and kinetics of IgM and IgG in the progression of COVID-19**. BMC Immunol. 17 févr 2021;22(1):14. | Qutob N, Awartani F, Salah Z, Asia M, Abu Khader I, Herzallah K, et al. **Seroprevalence of SARS-CoV-2 in the West Bank region of Palestine: a cross-sectional seroepidemiological study.** Bmj Open. 2021;11(2): e044552. | Richard A, Wisniak A, Perez-Saez J, Garrison-Desany H, Petrovic D, Piumatti G, et al. **Seroprevalence of anti-SARS-CoV-2 IgG antibodies, risk factors for infection and associated symptoms in Geneva, Switzerland: a population-based study** [Internet]. 2020 Dec [cited 2022 Jan 20] p. 2020.12.16.20248180. Available from: https://www.medrxiv.org/content/10.1101/2020.12.16.20248180v1 | Rosenberg ES, Tesoriero JM, Rosenthal EM, Chung R, Barranco MA, Styer LM, et al. **Cumulative incidence and diagnosis of SARS-CoV-2 infection in New York**. Ann Epidemiol. août 2020;48:23-29.e4. |
| LOCATION/ COUNTRY | China (in 4 countries) | West Bank, Palestine | Geneva, Switzerland | New York, in 99 grocery shops in 26 counties |
| SELECTION CRITERIA | Men and women, all ages | Men and women, all ages | Men and women, 5–94 years old, with asymptomatic cases | Men and women, all ages |
| TYPE OF POPULATIONS | Suspected patients, confirmed patients and consecutive follow-up patients | Doing housework in the West Bank, people visiting Palestinian medical laboratories | Population of the canton of Geneva were invited to participate in a seroprevalence study, along with household members five years and older (General population) | General population |
| SAMPLE SIZE | 571 were enrolled in the cross-sectional study, including 235 COVID-19 patients and 336 suspected | 1,355from 11 governorates, including 112 locations in the West Bank and 1136 peoples visiting Palestinian medical labs | 8,344 | 15,101 |
| SAMPLING METHOD | Random sampling | Random sampling | Random sampling | Random sampling |
| SEROPREVALENCE | Between 2.1 and 5.4% positives cases | The random sample of Palestinians living in the West Bank yielded 0% seroprevalence with 95% and an adjusted CI of 0.0043%. seroprevalence with 95% and adjusted CI (0% to 0.0043%), while the laboratory reference samples yielded an estimated seroprevalence of 0.354% with 95% and an adjusted seroprevalence of 0.354% with 95% and an adjusted CI (0.001325% to 0.011566%) | First study months: 7.1% (95% CrI 5.5–8.7) Second months: 9.0% (95% CrI 7.5–10.5) Third months: 7.1% (95% CrI 5.7–8.5) | 14% 2.139 positive cases including 300 adults |
| STUDY PERIOD | January-March 2020 | June-2020 | 6.April-30. June 2020 | April-2020 |
| RISK FACTORS | | | Age, gender, employment status, occupational category, change in work, specific change in work, educational level, single, smocking, BMI category, chronic health condition, risk-contact exposure | |

(*Continued*)

**Table 1.** (Continued)

| NUMBER | 1. | 2. | 3. | 4. |
|---|---|---|---|---|
| SYMPTOMS | Mild symptoms | | Fatigue, headache, sneezing/rhinorrhea, fever, cough, anosmia/disgeusia, myalgia/arthralgia, sore throat, dyspnea, loss of appetite, diarrhea, abdominal pain, nausea/vomiting, other symptoms, Asymptomatic cases | |
| NUMBER | 73. | 74. | 75. | 76. |
| REFERENCE | Rostami A., Sepidarkish M., M.G. Leeflang M., Mohammad Riahi S., Nourollahpour Shiadeh M., Esfandyari S., et al. **"SARS-CoV-2 seroprevalance worldwide:a systematic reviewand meta-analysis".** Clinical Microbiology and Infection. 26 Dec 2021. 27 (3):331–340. | Roxhed N, Bendes A, Dale M. **A translational multiplex serology approach to profile the prevalence of anti-SARS-CoV-2 antibodies in home-sampled blood**. MedRxiv 2020. | Samore MH, Looney A, Orleans B, Greene T, Seegert N, Delgado JC, et al. **SARS-CoV-2 seroprevalence and detection fraction in Utah urban populations from a probability-based sample** [Internet]. 2020 Oct [cited 2022 Jan 20] p. 2020.10.26.20219907. Available from: https://www.medrxiv.org/content/10.1101/2020.10.26.20219907v1 | Santarelli A, Lalitsasivimol D, Bartholomew N, Reid S, Reid J, Lyon C, et al. **The seroprevalence of SARS-CoV-2 in a rural southwest community**. J Osteopath Med. 1 févr 2021;121 (2):199-210. |
| LOCATION/COUNTRY | Worldwide | Stockholm, Sweden | Utah, Salt, Lake, Davis, and Summit, United States | Kingman area, Arizona |
| SELECTION CRITERIA | Men and women all ages | Men and women, 20–74 years old | Men and women, all ages | Men and women, 18 years and over, 380 women, 186 men, 458 white, 303 health professionals |
| TYPE OF POPULATIONS | General population | General population | General population, population of the four counties included in this serological survey | Community member and health care workers |
| SAMPLE SIZE | 399,265 | 2,000 | 8,108, 5,125 household | 566 took part in the final analysis |
| SAMPLING METHOD | Several techniques | Random households in urban Stockholm | Random sampling | Random sampling |
| SEROPREVALENCE | Between 0.37% to 22,1% positive cases<br>Northern Europe (5.27%)<br>Southern Europe (4.41%)<br>North America (4.41%)<br>Western Europe (3.17%)<br>East Asia (2.02%)<br>South America (1.45%) | 10.1–10.8% | The overall prevalence of IgG antibody to SARS-CoV-2 was estimated at 0.8%. The estimated seroprevalence-to-case count ratio was 2.4, corresponding to a detection fraction of 42% | The seroprevalence of SARS-CoV-2 was found to be 8.0% (45 of 566) in the sample |
| STUDY PERIOD | January-August 2020 | April-2020 | 4.May-30 June 2020 | September-October 2020 |
| RISK FACTORS | Ethnicity | Sex, age, flu-like | Location, children's household, gender, age, ethnicity, race, co-morbidities, exposure | |
| SYMPTOMS | | Breathing symptoms | Nasopharyngeal | Anosmia/ageusia, cough, chest congestion, fever, shortness of breath, chest pain |
| NUMBER | 77. | 78. | 79. | 80. |
| REFERENCE | Satpati PS, Sarangi SS, Gantait KS, Endow S, Mandal NC, Panchanan K, et al. **Sero-surveillance (IgG) of SARS-CoV-2 among Asymptomatic General population of Paschim Medinipur, West Bengal, India** [Internet]. medRxiv; 2020 [cited 2022 Feb 9]. p. 2020.09.12.20193219. Available from: https://www.medrxiv.org/content/10.1101/2020.09.12.20193219v1 | Shakiba M, Nazari SSH, Mehrabian F, Rezvani SM, Ghasempour Z, Heidarzadeh A. **Seroprevalence of COVID-19 virus infection in Guilan province, Iran**. medRxiv 2020; published online May 1. DOI:10.1101/2020.04.26.20079244 (preprint). | Sharma N, Sharma P, Basu S, Saxena S, Chawla R, Dushyant K, et al. **The seroprevalence and trends of SARS-CoV-2 in Delhi, India: A repeated population-based seroepidemiological study** [Internet]. 2020 Dec [cited 2022 Jan 20] p. 2020.12.13.20248123. Available from: https://www.medrxiv.org/content/10.1101/2020.12.13.20248123v1 | Shaw JA, Meiring M, Cummins T, Chegou NN, Claassen C, Du Plessis N, et al. **Higher SARS-CoV-2 seroprevalence in workers with lower socioeconomic status in Cape Town, South Africa**. Plos One. 25 févr 2021;16(2):e0247852. |

(*Continued*)

**Table 1.** (Continued)

| NUMBER | 1. | 2. | 3. | 4. |
|---|---|---|---|---|
| LOCATION/ COUNTRY | Paschim Medinipur, West Bengal, India | Guilan, Iran | Delhi, India | South Africa |
| SELECTION CRITERIA | Men and women, all ages | Men and women, all ages, | Men and women, 5 years old and over | Representation of all strata |
| TYPE OF POPULATIONS | Asymptomatic general population and RTPCR positive cases in 30 villages or wards of municipalities General population | General population (telephone invitation) | Age ≥5 years and residents of Delhi for at least the past six months | The workforce of a popular commercial and tourist complex |
| SAMPLE SIZE | 488 | 196 households, including 551 subjects | 47,470 | 405 volunteers |
| SAMPLING METHOD | Random sampling | Random cluster sampling | Multi-stage random sampling | Random sampling |
| SEROPREVALENCE | Of the 458 asymptomatic general population, 19 asymptomatic people found to be seropositive IgG for SARS-CoV-2 with Mean or average total seropositivity rate of 4.15%. 19 Out of 30 (63.33%) RTPCR positive patients found Seronegative Highest Seropositivity percentage found in Ghatal Municipality of 12.50% followed by Daspur II of 9.78%, Daspur I of 4.00% and Kharagpur Municipality of 3.70% and Midnapur Municipality of 6.25% | 22% | A total of 4267 (n = 15046), 4311 (n = 17409), and 3829 (n = 15015) positive tests indicatives of the presence of IgG antibody to SARS-CoV-2 were observed during the August, September, and October 2020 serosurvey rounds, respectively. The adjusted seroprevalence: August: 39% (95% CI 27.65–29.14) September: 24.08% (95% CI 23.43–24.74) October: 24.71% (95% CI 24.01, 25.42%) | 96 (23.7%) were positive for SARS-CoV-2 IgG Of those who tested positive 46 (47.9%) reported no symptoms of COVID-19 in the previous 6 months The specificity of the test was 98.54% (95%CI 94.82%-99.82%) in the pre-COVID controls |
| STUDY PERIOD | July-August 2020 | April-2020 | August-October 2020 | August-September 2020 |
| RISK FACTORS | Gender, age, locality, slum area, employment, education, statut social, travel history, hydroxychloroquine prophylaxis, migrant labourer, history of chicken pox, inluenza vaccination status. | | Age, gender, education, household size, income, living in Urban Slum, location | |
| SYMPTOMS | Asymptomatic cases, | General, respiratory or gastrointestinal symptoms. | | New cough, fever or chills, new dyspnoea, sore throat, loss of smell, diarrhoea, nausea and/or vomiting |
| NUMBER | 81. | 82. | 83. | 84. |
| REFERENCE | Sherman A, Reuben J, David N, Quasie-Woode DP, Gunn JKL, Nielsen CF, et al. **SARS-CoV-2 Seroprevalence Survey Among District Residents Presenting for Serologic Testing at Three Community-Based Test Sites—Washington, DC, July–August, 2020** [Internet]. 2021 Feb [cited 2022 Jan 20] p. 2021.02.15.21251764. Available from: https://www.medrxiv.org/content/10.1101/2021.02.15.21251764v1 | Silva AAM da, Lima-Neto LG, Azevedo C de MP e S de, Costa LMM da, Bragança MLBM, Filho AKDB, et al. **Population-based seroprevalence of SARS-CoV-2 is more than halfway through the herd immunity threshold in the State of Maranhão, Brazil** [Internet]. medRxiv; 2020 [cited 2022 Feb 2]. p. 2020.08.28.20180463. Available from: https://www.medrxiv.org/content/10.1101/2020.08.28.20180463v1 | Silveira MF, Barros AJD, Horta BL, Pellanda LC, Dellagostin OA, Struchiner CJ, et al. **Repeated population-based surveys of antibodies against SARS-CoV-2 in Southern Brazil** [Internet]. medRxiv; 2020 [cited 2022 Jan 31]. p. 2020.05.01.20087205. Available from: https://www.medrxiv.org/content/10.1101/2020.05.01.20087205v2 | Snoeck CJ, Vaillant M, Abdelrahman T, Satagopam VP, Turner JD, Beaumont K, et al. **Prevalence of SARS-CoV-2 infection in the Luxembourgish population–the CON-VINCE study** [Internet]. 2020 May [cited 2022 Jan 18] p. 2020.05.11.20092916. Available from: https://www.medrxiv.org/content/10.1101/2020.05.11.20092916v1 |
| LOCATION/ COUNTRY | Washington, DC, United States | State of Moranhao, Brazil | Southern Brazil | Luxembourg |
| SELECTION CRITERIA | Men and women, 6 years old and over | Men and women, between 1 and over 70 years old, with asymptomatic cases | Household members living in one of the nine sentinel cities in Brazil | Men and women, 18 years old and over |
| TYPE OF POPULATIONS | DC Residents | A population-based household survey | General population | General population |

(*Continued*)

**Table 1.** (Continued)

| NUMBER | 1. | 2. | 3. | 4. |
|---|---|---|---|---|
| SAMPLE SIZE | 508 households and 671 samples | 3,156 | 4,188 in first round 4,500 in the second round | 1,862 |
| SAMPLING METHOD | Random sampling | A three-stage cluster sampling stratified by four state regions was used | Multistage sampling to select 50 census tracts with probability proportionate to size in each sentinel city, and 10 households at random in each tract based on census listings updated in 2019 One individual randomly selected through household | Random sampling |
| SEROPREVALENCE | 7.6%. | Seroprevalence of total antibodies against SARS-CoV-2 was 40·4% (95%CI 35·6–45·3) | In the first round: 0.0477%; 95% confidence interval (CI) 0.0058%;0.1724%) In the second round: 0.1333%; 95% CI 0.0489%;0.2900% | IgA: 11.07 (95%CI = [9.54;12.60]) IgG: 2.09% (95%CI = [1.37;2.82]) |
| STUDY PERIOD | 27. July-21. August 2020 | 27. July-8. August 2020 | 28. Feb- 30. Apr 2020 | 15. April- 5. May 2020 |
| RISK FACTORS | Ethnicity/Race, age, work | Age, location, ethnicity/race, number of residents, social status | The most densely populated centres were the most affected by COVID-19 | / |
| SYMPTOMS | | 62·2% had more than three symptoms, 11·1% had one or two symptoms, and 26·0% were asymptomatic We also showed that among the infected, 26·0% were asymptomatic and 11·1% had one or two symptoms and that the predominant symptoms among those who tested positive for SARS-CoV-2 were anosmia/hyposmia (49·5%), ageusia/dysgeusia (47·7%), fever (45·6%), headache (45·4%), myalgia (43·6%), and fatigue (41·1%). The infection fatality rate was 0.17%, higher for males and advanced age groups | | |
| NUMBER | 85. | 86. | 87. | 88. |
| REFERENCE | Snyder T, Ravenhurst J, Cramer EY, Reich NG, Balzer LB, Alfandari D, et al. **Serological surveys to estimate cumulative incidence of SARS-CoV-2 infection in adults (Sero-MAss study), Massachusetts, July-August 2020: a mail-based cross-sectional study.** MedRxiv Prepr Serv Health Sci. 9 mars 2021; | Sood N, Simon P, Ebner P, Eichner D, Reynolds J, Bendavid E, et al. **Seroprevalence of SARS-CoV-2-Specific Antibodies Among Adults in Los Angeles County, California, on April 10–11, 2020.** JAMA. 16 juin 2020;323(23):2425-7. | Soriano V, Meiriño R, Corral O, Guallar MP. **Severe Acute Respiratory Syndrome Coronavirus 2 Antibodies in Adults in Madrid, Spain.** Clin Infect Dis Off Publ Infect Dis Soc Am. 15 mars 2021;72(6):1101-2. | Stout RL, Rigatti SJ. **The Silent Pandemic COVID-19 in the Asymptomatic Population** [Internet]. 2021 Jan [cited 2022 Jan 20] p. 2020.11.10.20215145. Available from: https://www.medrxiv.org/content/10.1101/2020.11.10.20215145v3 |
| LOCATION/ COUNTRY | Massachusetts | Los Angeles County, California | Madrid, Spain | United States |
| SELECTION CRITERIA | Men and women, 18 years old and over | Men and women, 18 years old and over, 60% female, 55% between 35 and 54 years old, 58% white | Men and women, asymptomatic adults | Life insurance applicants who had blood tests performed as part of life insurance underwriting at Clinical Reference Laboratories |
| TYPE OF POPULATIONS | Students, professors, librarians and staff of the University of Massachusetts at Amherst, with no previous covid-19 diagnosis, were invited to participate in this study with a member of their household | General population | Population attending the Clinical University of Madrid | Asymptomatic general population |
| SAMPLE SIZE | 762 | 865 | 674 | 126,587 |

*(Continued)*

**Table 1.** (Continued)

| NUMBER | 1. | 2. | 3. | 4. |
|---|---|---|---|---|
| SAMPLING METHOD | Random sampling | Random sampling, they used a representative representative property database of the county managed by LRW Group, a market research company to select participants. | Random sampling | One fifth of all samples tested from a pool of life insurance applicants |
| SEROPREVALENCE | 5,3% and 4% | 35% out of 865 positive tests i.e. 4.06% | 13,80% | May-June, September, and December timeframes: 3.0%, 6.6% and 10.4%, respectively |
| STUDY PERIOD | July-August 2020 | April-2020 | April-May 2020 | May-December 2020 |
| RISK FACTORS | | Age, kids | The most densely populated centres were the most affected by COVID-19 | Age |
| SYMPTOMS | | Fever, cough shortness of breath, loss of sense of smell or taste | Mild symptoms, asymptomatic cases | Asymptomatic cases |
| NUMBER | 89. | 90. | 91. | 92. |
| REFERENCE | Stringhini S, Wisniak A, Piumatti G, Azman AS, Lauer SA, Baysson H, et al. **Seroprevalence of anti-SARS-CoV-2 IgG antibodies in Geneva, Switzerland (SEROCoV-POP): a population-based study.** Lancet Lond Engl. 1 août 2020;396 (10247):313-9. | Sulcebe G, Ylli A, Cenko F, Kurti-Prifti M. **Rapid increase of SARS-CoV-2 seroprevalence during the 2020 pandemic year in the population of the city of Tirana, Albania** [Internet]. 2021 Feb [cited 2022 Jan 20] p. 2021.02.18.21251776. Available from: https://www.medrxiv.org/content/10.1101/2021.02.18.21251776v1 | Sutton M, Cieslak P, Linder M. **Notes from the Field: Seroprevalence Estimates of SARS-CoV-2 Infection in Convenience Sample**—Oregon, May 11-June 15, 2020. MMWR Morb Mortal Wkly Rep. 14 août 2020;69(32):1100-1. | Tabanejad Z, Darvish S, Boroujeni ZB, Asadi SS, Mesri M, Raiesi O, et al. **Seroepidemiological Study of Novel Corona Virus (CoVID-19) in Tehran, Iran** [Internet]. medRxiv; 2021 [cited 2022 Feb 9]. p. 2021.01.18.20248911. Available from: https://www.medrxiv.org/content/10.1101/2021.01.18.20248911v1 |
| LOCATION/ COUNTRY | Geneva, Switzerland | city of Tirana, Albania | Oregon, California | Teheran, Iran |
| SELECTION CRITERIA | Men and women, all ages | Men and women, individuals 20–70 years old | Men and women, 18 years old and over | Men and women, all ages |
| TYPE OF POPULATIONS | Former participants in the Health Bus study and members of their household | General population (communities) | Sample visiting an ambulatory centre, emergency and hospital between May and June 2020 | Patients for Valiasr, Sajad and Ghaem hospitals, Tehran, the capital of Iran |
| SAMPLE SIZE | 2,766samples from 1,339 households | 1,081 | 897 | 1,375 |
| SAMPLING METHOD | Random sampling | In two rounds using two independently selected samples randomly selected from lists of the inhabitants | Random sampling | Random sampling |
| SEROPREVALENCE | 8–5% in the second week, 10–9% in the third week, 6–6% in the fourth week, and 10–8% in the fifth week | In early July: 7.5% (95% CI: 4.3%-10.7%) Late December 2020: 48.2% (95% CI: 44.8%-51.7%) | 1%, 9 positive cases out of 897 samples, 0% seroprevalence in children under 17 | Among all participants, 291 patients (21.2%) were positive for either IgM or IgG antibodies, indicating past or present infection (P <0.05) |
| STUDY PERIOD | April-May 2020 | July-December 2020 | May-June 2020 | April-October 2020 |
| RISK FACTORS | | | | Co-morbidities, gender, age |
| SYMPTOMS | | Reported symptoms | | |
| NUMBER | 93. | 94. | 95. | 96. |

(*Continued*)

**Table 1.** (Continued)

| NUMBER | 1. | 2. | 3. | 4. |
|---|---|---|---|---|
| REFERENCE | Takita M, Matsumura T, Yamamoto K, Yamashita E, Hosoda K, Hamaki T, et al. **Geographical Profiles of COVID-19 Outbreak in Tokyo: An Analysis of the Primary Care Clinic-Based Point-of-Care Antibody** Testing. J Prim Care Community Health. déc 2020;11:2150132720942695. | Tess BH, Granato CFH, Alves MCGP, Pintao MC, Rizzatti E, Nunes MC, et al. **SARS-CoV-2 seroprevalence in the municipality of São Paulo, Brazil, ten weeks after the first reported case** [Internet]. medRxiv; 2020 [cited 2022 Feb 2]. p. 2020.06.29.20142331. Available from: https://www.medrxiv.org/content/10.1101/2020.06.29.20142331v1 | Tsertsvadze T, Gatserelia L, Mirziashvili M, Dvali N, Abutidze A, Metchurtchlishvili R, et al. **SARS-CoV-2 antibody seroprevalence in Tbilisi, the capital city of country of Georgia** [Internet]. 2020 Sep [cited 2022 Jan 20] p. 2020.09.18.20195024. Available from: https://www.medrxiv.org/content/10.1101/2020.09.18.20195024v1 | Velumani A, Nikam C, Suraweera W, Fu SH, Gelband H, Brown P, et al. **SARS-CoV-2 Seroprevalence in 12 Cities of India from July-December 2020** [Internet]. medRxiv; 2021 [cited 2022 Jan 31]. p. 2021.03.19.21253429. Available from: https://www.medrxiv.org/content/10.1101/2021.03.19.21253429v1 |
| LOCATION/ COUNTRY | Tokyo, Japan | Sao Paulo, Brazil | Tbilisi, Georgia | 12 Cities, India |
| SELECTION CRITERIA | Men and women, all ages | Men and women, 18 years old and over | Men and women, 18–64 years old | People of all ages who underwent testing for SARS-CoV-2 antibodies between June 12, 2020 and December 31, 2020 by Thyrocare Laboratories |
| TYPE OF POPULATIONS | Asymptomatic patients Patients in the two community clinics in Tokyo (Navitas Clinic Shinjuku and Tachikawa) | A population-based household survey, who are residents of six districts in São Paulo City. | Residents of capital city of Tbilisi | Urban population |
| SAMPLE SIZE | 1,071 | 517 (299 randomly-selected adults and 218 cohabitants | 1,068 | 448,518 |
| SAMPLING METHOD | Random sampling, from web publication on the homepage of the clinics | Random sampling | Convenience sampling | Based on geographical areas and laboratories Random sampling |
| SEROPREVALENCE | 4.68% positive cases in Tokyo for 1.83%in the other regions | 4.7% (95% CI 3.0–6.6%) | Nine persons tested positive for IgG: crude seroprevalence: 0.84%, (95% CI: 0.33%-1.59%), weighted seroprevalence: 0.94% (95% CI: 0.37%-1.95%), weighted and adjusted for test accuracy: 1.02% (95% CI: 0.38%-2.18%). The seroprevalence estimates translate into 7,200 to 8.800 infections among adult residents of Tbilisi | 31% (140,631) |
| STUDY PERIOD | April-May 2020 | May-2020 | May-2020 | July-December 2020 |
| RISK FACTORS | Illness cohabitation, co-worker, working environment, age, location | Age, gender, race, education, | | |
| SYMPTOMS | Fever, asymptomatic cases | Fever, couch, shortness of breath, sore throat, rhinorrhea, fatigue, myalgia, diarrhea, ageusia, anosmia | Fever, cough, shortness of breath, fatigue, sore throat, rhinorrhea, loss of smell/taste | |
| NUMBER | 97. | 98. | 99. | 100. |
| REFERENCE | Vena A, Berruti M, Adessi A, Blumetti P, Brignole M, Colognato R, et al. **Prevalence of Antibodies to SARS-CoV-2 in Italian Adults and Associated Risk Factors.** J Clin Med. 27 août 2020;9(9). | Vodičar PM, Valenčak AO, Zupan B, Županc TA, Kurdija S, Korva M, et al. **Low prevalence of active COVID-19 in Slovenia: a nationwide population study of a probability-based sample**. Clin Microbiol Infect. 2020 Nov 1;26 (11):1514–9. | Vu SL, Jones G, Anna F, Rose T, Richard J-B, Bernard-Stoecklin S, et al. **Prevalence of SARS-CoV-2 antibodies in France: results from nationwide serological surveillance** [Internet]. 2020 Oct [cited 2022 Jan 20] p. 2020.10.20.20213116. Available from: https://www.medrxiv.org/content/10.1101/2020.10.20.20213116v1 | Wang X, Gao W, Cui S, Zhang Y, Zheng K, Ke J, et al**. A population-based seroprevalence survey of severe acute respiratory syndrome coronavirus** 2 infection in Beijing, China [Internet]. medRxiv; 2020 [cited 2022 Feb 9]. p. 2020.09.23.20197756. Available from: https://www.medrxiv.org/content/10.1101/2020.09.23.20197756v1 |
| LOCATION/ COUNTRY | In 5 administrative departments of the regions of Liguria and Lombardie in Italy | Slovenia | France | Beijing, China |

(*Continued*)

**Table 1.** (Continued)

| NUMBER | 1. | 2. | 3. | 4. |
|---|---|---|---|---|
| SELECTION CRITERIA | Men and women, 18 years old and over | Men and women, all ages | Men and women, all ages | Men and women, residents who aged > 1 year old |
| TYPE OF POPULATIONS | A large sample of healthy adult volunteers, not hospitalized | All permanent and temporary residents of Slovenia | French on the national territoy and people who are living in Guadeloupe, Martinique, Mayotte, French Guiana, La Réunion | General population (Residents) |
| SAMPLE SIZE | 3,609 | 1,368 | 11,021 | 2,184 |
| SAMPLING METHOD | Random sampling | Random sampling | Random sampling | Multi-stage cluster random sampling |
| SEROPREVALENCE | An average seroprevalence of 11% with 9.2% of men, 12.5% of women and 13.5% of the over-55s That is 398 positive cases out of 3609 | Two of 1366 participants tested positive for SARS-CoV-2 RNA (prevalence 0.15%; posterior mean 0.18%, 95% Bayesian confidence interval 0.03e0.47; 95% highest density region (HDR) 0.01e0.41) | Mid-March: 0.41% [0.05−0.88] Mid-April: 4.14% [3.31−4.99] Mid-May: 4.93% [4.02−5.89] | The seroprevalence of COVID-19 in Beijing was estimated < 0.17% |
| STUDY PERIOD | March-April 2020 | March-October 2020 | March-May 2020 | April 2020 |
| RISK FACTORS | | | Age, location | |
| SYMPTOMS | Influenza-like illness or loss of sense of smell or taste were | Acute respiratory symptoms and/ or fever | | Mild symptoms |
| NUMBER | 101 | 102 | 103 | 104 |
| REFERENCE | Ward H, Cooke G, Atchison C, Whitaker M, Elliott J, Moshe M, et al. **Declining prevalence of antibody positivity to SARS-CoV-2: a community study of 365,000 adults** [Internet]. 2020 Oct [cited 2022 Jan 20] p. 2020.10.26.20219725. Available from: https://www.medrxiv.org/content/10.1101/2020.10.26.20219725v1 | Ward H, Atchison C, Whitaker M, Ainslie KE, Elliott J, Okell L, et al. **Antibody prevalence for SARS-CoV-2 following the peak of the pandemic in England: REACT2 study in 100,000 adults** [Internet]. 2020 Aug [cited 2022 Jan 20] p. 2020.08.12.20173690. Available from: https://www.medrxiv.org/content/10.1101/2020.08.12.20173690v2 | Waterfield T, Watson C, Moore R, Ferris K, Tonry C, Watt AP, et al. **Seroprevalence of SARS-CoV-2 antibodies in children—A prospective multicentre cohort study** [Internet]. 2020 Sep [cited 2022 Jan 20] p. 2020.08.31.20183095. Available from: https://www.medrxiv.org/content/10.1101/2020.08.31.20183095v1 | Weis S, Scherag A, Baier M, Kiehntopf M, Kamradt T, Kolanos S, et al. **Seroprevalence of SARS-CoV-2 antibodies in an entirely PCR-sampled and quarantined community after a COVID-19 outbreak-the CoNAN study**. medRxiv 2020; published online July 17. DOI:10.1101/2020.07.15.20154112 (preprint). |
| LOCATION/ COUNTRY | United Kingdom, England | England | 5 United Kingdom sites (Belfast, Cardiff, Manchester, Glasgow and London) | Neustadt-am-Rennsteig, Germany |
| SELECTION CRITERIA | Men and women, 18 years old and over | Men and women, 18 years and over | Men and women, 2–15 years old | Men and women, all ages, |
| TYPE OF POPULATIONS | Adult population of England (General population) | General population (representative sample) | Children of healthcare workers | All households in the community of Neustadt-am-Rennsteig were informed (by mail prior) General population |
| SAMPLE SIZE | 99,908, 105,829 and 159,367 | 109,076 | 992 | 626 |
| SAMPLING METHOD | Random sampling | Random sampling | Random sampling | Random sampling |
| SEROPREVALENCE | Antibody prevalence, adjusted for test characteristics and weighted to the adult population of England, declined from 6.0% [5.8, 6.1], to 4.8% [4.7, 5.0] and 4.4% [4.3, 4.5], a fall of 26.5% [-29.0, -23.8] over the three months of the study. There was a decline between rounds 1 and 3 in all age groups, with the highest prevalence of a positive result and smallest overall decline in positivity in the youngest age group (18–24 years: -14.9% [-21.6, -8.1]), and lowest prevalence and largest decline in the oldest group (75+ years: -39.0% [-50.8, -27.2]) The decline from rounds 1 to 3 was largest in those who did not report a history of COVID19, (-64.0% [-75.6, -52.3]), compared to -22.3% ([-27.0, -17.7]) in those with SARS-CoV-2 infection confirmed on PCR | Adjusted and re-weighted: 6.0% (95% Cl: 5.8, 6.1) | There were 68 (6.9%) participants with positive SARS-CoV-2 antibody tests indicative of previous SARS-CoV-2 infection Belfast had significantly lower seroprevalence than all other sites at 0.9% (95% CI 0.2 to 3.3, n = 215) | Only 8·4% of the tested population were seropositive for anti-SARS-CoV-2 antibodies in which 6·2% (38/610) had proven SARS-CoV-2 infection, indicating 22 a low rate of asymptomatic cases |
| STUDY PERIOD | June-September 2020 | 20. June-13. July 2020 | 16. April-3. July 2020 | May-2020 |

(*Continued*)

**Table 1.** (Continued)

| NUMBER | 1. | 2. | 3. | 4. |
|---|---|---|---|---|
| RISK FACTORS | Location, age, gender, employment, household size, population density | Employment (Heatlth and care home worker), ethnicity, age, smoking status (no smokers), household size and deprivation quintile (more deprived ares or living in larger households), BMI (overweight or obese), region, symptoms severity and contact with case (having more severe symptoms, and having contact with a confirmed or suspected case) | Known household, the role of the parent, contact with a household member | |
| SYMPTOMS | No symptoms, atypical symptoms only, screening symptoms | Severe symptoms | Fever 21/68 (31%), gastrointestinal symptoms (diarrhoea, vomiting and abdominal cramps) 13/68 (19%) and headache 12/68 (18%). The presence of fever, cough or changes in a sense of smell/taste were recorded in 26/68 (38%) of participants | |
| NUMBER | 105 | 106 | 107 | 108 |
| REFERENCE | Wells PM, Doores KM, Couvreur S. **Estimates of the rate of infection and asymptomatic COVID-19 disease in a population sample from se England.** MedRxiv 2020. | Wiens, Kirsten W, and al. **"Seroprevalence of anti-SARS-CoV-2 IgG antibodies in Juba, South Sudan: a population-based study".** medRxiv. 2021.doi:10.1101/2021.03.08.21253009 | Williamson JC, Wierzba TF, Santacatterina M, Munawar I, Seals AL, Ballard CAP, et al. **Analysis of Accumulated SARS-CoV-2 Seroconversion in North Carolina: The COVID-19 Community Research Partnership** [Internet]. 2021 Mar [cited 2022 Jan 20] p. 2021.03.11.21253226. Available from: https://www.medrxiv.org/content/10.1101/2021.03.11.21253226v1 | Zejda JE, Brożek GM, Kowalska M, Barański K, Kaleta-Pilarska A, Nowakowski A, et al. **Seroprevalence of Anti-SARS-CoV-2 Antibodies in a Random Sample of Inhabitants of the Katowice Region, Poland.** Int J Environ Res Public Health. 19 mars 2021;18(6). |
| LOCATION/ COUNTRY | England | Juba, South Sudan | North California, United States | Katowice, Poland |
| SELECTION CRITERIA | Men and women, 19–86 years old with asymptomatic cases | Men and women, all ages | Men and women, 18 years old and over | Men and women, all ages |
| TYPE OF POPULATIONS | General population | General population | Adult community residents (General population) | General population |
| SAMPLE SIZE | 431 | 2,214 | 17,688 | 1,167 |
| SAMPLING METHOD | Random sampling | Random sampling | Random sampling | Random sampling |
| SEROPREVALENCE | 12% | 22.3% had anti-SRAS-CoV-2 IgG titers above the levels of the pre-malignant samples | The average number of serology test results submitted per participant was 3.0 (±1.9). At December 20, 2020, the overall probability of seropositivity in the CCRP population was 32.6%. At February 15, 2021 the probability among non-healthcare workers was 49% | The prevalence of IgG seropositivity was 11.4% (95% CI 9.5–13.2%). (95% CI: 9.5–13.2%) and IgM seropositivity was 4.6 (95% CI: 3.5–5.8%). |
| STUDY PERIOD | March-June 2020 | August-September 2020 | April 2020-February 2021 | March-November 2020 |
| RISK FACTORS | | | | |
| SYMPTOMS | Asympatomatic cases, neither fever, persistent cough, nor anosmia | | | Fever, chills fatigue, cough, clogged nose, dyspnea/trouble breathing, headache, nausea and loss of smell/taste |
| NUMBER | 109 | 110 | 111 | 112 |

(Continued)

**Table 1.** (Continued)

| NUMBER | 1. | 2. | 3. | 4. |
|---|---|---|---|---|
| REFERENCE | Adetifa IMO, Uyoga S, Gitonga JN, Mugo D, Otiende M, Nyagwange J, et al. **Temporal trends of SARS-CoV-2 seroprevalence in transfusion blood donors during the first wave of the COVID-19 epidemic in Kenya** [Internet]. 2021 Feb [cited 2022 Jan 20] p. 2021.02.09.21251404. Available from: https://www.medrxiv.org/content/10.1101/2021.02.09.21251404v1 | Alandijany TA, El-Kafrawy SA, Al-Ghamdi AA, Qashqari FS, Faizo AA, Tolah AM, et al. **Lack of Antibodies to SARS-CoV-2 among Blood Donors during COVID-19 Lockdown: A Study from Saudi Arabia.** Healthcare. Janv 2021;9(1):51. | Amorim Filho L, Szwarcwald CL, Mateos S de OG, Leon ACMP de, Medronho R de A, Veloso VG, et al. **Seroprevalence of anti-SARS-CoV-2 among blood donors in Rio de Janeiro, Brazil.** Rev Saude Publica. 2020;54:69. | Bendavid E, Mulaney B, Sood N, Shah S, Bromley-Dulfano R, Lai C, et al. **COVID-19 antibody seroprevalence in Santa Clara County, California.** Int J Epidemiol. 22 févr 2021; |
| LOCATION/ COUNTRY | Kenya, Africa | Saudi Arabia | Rio de Janeiro, Brazil | Santa Clara County, California |
| SELECTION CRITERIA | Men and women, aged 16–65 years, weighing ≥50kg, with haemoglobin of 12·5g/dl, a normal blood pressure (systolic 120–129 mmHg and diastolic BP of 80–89 mmHg), a pulse rate of 60–100 beats per minute and without any history of illness in the past 6 months. | Men and women, all ages | All Voluntary blood donors of the Hemorio, Brazil | Individual who could be tested (obtain blood or blood clotted), living in Santa Clara County and tests results matched with personal data |
| TYPE OF POPULATIONS | Blood donors | Healthy blood donors attending one of the largest hospitals in the western region of Saudi Arabia | Voluntary blood donors | Adults and children (county resident) |
| SAMPLE SIZE | 9,922 | 956 | 2,857 | 3,330 |
| SAMPLING METHOD | Random sampling | Random sampling | Volunteers and adjusted for sex and age group Random sampling | Using Facebook ads targeting a sample of individuals living within the county by demographic and geographic characteristics |
| SEROPREVALENCE | Period 1: 30. April-19. June = the adjusted seroprevalence of SARS-CoV-2 was 5.2% (95% CI 3.7–6.7%). Period 2: 20. June-19. August = it had risen to 9.1% (95% CI 7.2–11.3%) Period 3: 20. August-30. September = it was maintained at 9.1% (95% CI 7.6–10.8% | 14 positive cases out of 956 i.e. 0.27% | Unadjusted 4.0% (95% CI: 3.3–4.7), weighted by Rio de Janeiro State Population 3.8% (95% CI: 3.1–4.5) | Raw prevalence: 1.5% (exact binomial 95CI 1.1–2.0%) |
| STUDY PERIOD | April-September 2020 | January-May 2020 | 14.April-27, April 2020 | 3. April-4. April 2020 |
| RISK FACTORS | Age, sex and residence characteristics | Tabagism | Period of blood collection, age, level of education | The most densely populated centres were the most affected by COVID-19 Age, Young people (between 18–30 years old), low level of education |
| SYMPTOMS | | | | Fever, cough, shortness of breath, runny nose, sore throat, loss of smell, loss of taste, no symptoms |
| NUMBER | 113 | 114 | 115 | 116 |
| REFERENCE | Buss LF, Prete CA, Abrahim CM, Mendrone A, Salomon T, Almeida-Neto C de, et al. **COVID-19 herd immunity in the Brazilian Amazon** [Internet]. medRxiv; 2020 [cited 2022 Feb 9]. p. 2020.09.16.20194787. Available from: https://www.medrxiv.org/content/10.1101/2020.09.16.20194787v1 | Chang L, Hou W, Zhao L, Zhang Y, Wang Y, Wu L, et al. **The prevalence of antibodies to SARS-CoV-2 among blood donors in China** [Internet]. 2020 Jul [cited 2022 Jan 18] p. 2020.07.13.20153106. Available from: https://www.medrxiv.org/content/10.1101/2020.07.13.20153106v1 | Davis G, York AJ, Bacon WC, Suh-Chin L, McNeal MM, Yarawsky AE, et al. **Seroprevalence of SARS-CoV-2 Infection in Cincinnati Ohio USA from August to December 2020** [Internet]. 2021 Mar [cited 2022 Jan 20] p. 2021.03.11.21253263. Available from: https://www.medrxiv.org/content/10.1101/2021.03.11.21253263v1 | Dopico XC, Muschiol S, Christian M, Hanke L, Sheward DJ, Grinberg NF, et al. **Seropositivity in blood donors and pregnant women during the first year of SARS-CoV-2 transmission in Stockholm, Sweden** [Internet]. medRxiv; 2021 [cited 2022 Jan 31]. p. 2020.12.24.20248821. Available from: https://www.medrxiv.org/content/10.1101/2020.12.24.20248821v2 |
| LOCATION/ COUNTRY | Brazilian Amazon | Wuhan, Shenzhen and Shijiazhuang, China | Cincinnati Ohio, USA | Stockholm, Sweden |

(*Continued*)

**Table 1.** (Continued)

| NUMBER | 1. | 2. | 3. | 4. |
|---|---|---|---|---|
| SELECTION CRITERIA | Men and women, blood donors living in Manaus or Sao Paulo. | Men and women, all blood donors living in Wuhan, Shenzhen, and Shijiazhuang | Men and women, volunteers and healthy unique blood donors presenting to the Hoxworth Blood Center | Men and women, asymptomatic blood donors and pregnant women |
| TYPE OF POPULATIONS | Blood donors | Healthy blood donors | Healthy blood donors | Healthy blood donors |
| SAMPLE SIZE | 1,000 | 38,144 | 9,550 | 5,100 |
| SAMPLING METHOD | Based on geographical areas Random sampling | Random sampling (all blood donors who lived in one of the three cities were enrolled in the study) | Random sampling | Random sampling |
| SEROPREVALENCE | In june: 44% | 2.29% (407/17,794, 95%CI: 2.08% to 2.52%) in Wuhan, 0.029% (2/6.810, 95%CI: 0.0081% to 0.11%) in Shenzhen, and 0.0074% (1/13.540, 95%CI: 0.0013% to 0.042%) in Shijiazhuang | 8.40% | 19.2% (95% Bayesian CI [15.1–24.4]) at the end of February 2021 |
| STUDY PERIOD | February-August 2020 | January-April 2020 | August-December 2020 | 14. March 2020-February 2021 |
| RISK FACTORS | | Period of blood donation, female and older age | Age, regions | |
| SYMPTOMS | | | | Asymptomatic cases |
| NUMBER | 117 | 118 | 119 | 120 |
| REFERENCE | Erikstrup C, Hother CE, Pedersen OBV, Mølbak K, Skov RL, Holm DK, et al. **Estimation of SARS-CoV-2 infection fatality rate by real-time antibody screening of blood donors** [Internet]. medRxiv; 2020 [cited 2022 Feb 9]. p. 2020.04.24.20075291. Available from: https://www.medrxiv.org/content/10.1101/2020.04.24.20075291v1 | Fiore JR, Centra M, De Carlo A, Granato T, Rosa A, Sarno M, et al. **FAR AWAY FROM HERD IMMUNITY TO SARS-CoV-2: results from a survey in healthy blood donors in South Eastern Italy** [Internet]. Infectious Diseases (except HIV/AIDS); 2020 Jun [cited 2022 Jan 18]. Available from: http://medrxiv.org/lookup/doi/10.1101/2020.06.17.20133678 | Fischer B, Knabbe C, Vollmer T. **SARS-CoV-2 IgG seroprevalence in blood donors located in three different federal states, Germany, March to June 2020.** Euro Surveill Bull Eur Sur Mal Transm Eur Commun Dis Bull. juill 2020;25(28). | Gallian P, Pastorino B, Morel P, Chiaroni J, Ninove L, de Lamballerie X. **Lower prevalence of antibodies neutralizing SARS-CoV-2 in group O French blood donors**. Antiviral Res. sept 2020;181:104880. |
| LOCATION/COUNTRY | Denmark | Apulia region, South Eastern Italy | Three federal states, German | France |
| SELECTION CRITERIA | Men and women, danish blood donors aged 17–69 years giving blood between April 6 to 17 2020 | Men and women, apparently healthy subjects, 18–65 years old | Men and women, all ages | Men and women, all ages, asymptomatic or pauci-symptomatic SARS-CoV-2 cases |
| TYPE OF POPULATIONS | Blood donors | Healthy blood donors | Regular blood donors | Blood donors |
| SAMPLE SIZE | 9,496 | 904 | 3,186 | 998 |
| SAMPLING METHOD | First donors were included Random sampling | Random sampling | Random sampling | Random sampling |
| SEROPREVALENCE | Combined adjusted: 1.7% (CI: 0.9–2.3) | 0,99% (9/304) | IgG seroprevalence was 0.91% (95% confidence interval (CI): 0.58–1.24) overall, ranging from 0.66% (95% CI: 0.13–1.19) in Hesse to 1.22% (95% CI: 0.33–2.10) in Lower Saxony | 2.82% for men and 2.69% for women. 1.32% for group O compared to 3.86% for the other groups |
| STUDY PERIOD | 6.April-17. April 2020 | 1. May-31. May 2020 | March-June 2020 | April-May 2020 |
| RISK FACTORS | The most densely populated centres were the most affected by COVID-19 Age, kids, presence of an seropositiv case in the household, region | | Presence of an seropositiv case in the household | The most densely populated centres were the most affected by COVID-19 Age, young people (between 18–30 years old) |

*(Continued)*

**Table 1.** (Continued)

| NUMBER | 1. | 2. | 3. | 4. |
|---|---|---|---|---|
| SYMPTOMS | | | Fever, leucocyte count | Asymptomatic or pauci-symptomatic SARS-CoV-2 infections |
| NUMBER | 121 | 122 | 123 | 124 |
| REFERENCE | Germain N, Herwegh S, Hatzfeld A-S, Bocket L, Prevost B, Danze P-M, et al. **Retrospective study of COVID-19 seroprevalence among tissue donors at the onset of the outbreak before implementation of strict lockdown measures in France.** Cell Tissue Bank. | Gidding HF, Machalek DA, Hendry AJ, Quinn HE, Vette K, Beard FH, et al. **Seroprevalence of SARS-CoV-2-specific antibodies in Sydney after the first epidemic wave of 2020.** Med J Aust. mars 2021;214(4):179-85. | Jin DK, Nesbitt DJ, Yang J, Chen H, Horowitz J, Jones M, et al. **Seroprevalence of Anti-SARS-CoV-2 Antibodies in a Cohort of New York City Metro Blood Donors using Multiple SARS-CoV-2 Serological Assays: Implications for Controlling the Epidemic and "Reopening"** [Internet]. 2020 Nov [cited 2022 Jan 20] p. 2020.11.06.20220087. Available from: https://www.medrxiv.org/content/10.1101/2020.11.06.20220087v1 | Kamath K, Baum-Jones E, Jordan G, Haynes W, Waitz R, Shon J, et al. **Prevalence of antibodies to SARS-CoV-2 in healthy blood donors in New York** [Internet]. 2020 Oct [cited 2022 Jan 20] p. 2020.10.19.20215368. Available from: https://www.medrxiv.org/content/10.1101/2020.10.19.20215368v1 |
| LOCATION/COUNTRY | France | Sydney | New-York City | New-York, United States |
| SELECTION CRITERIA | Men and women, all ages | Men and women, all ages Pregnant women aged 20–39, Australian between 20 and 69 | Men and women, 16–78 years old | Men and women, 17–80 years old, with asymptomatic cases |
| TYPE OF POPULATIONS | A tissue donor population | They had provided blood for testing in some of the diagnose pathology departments. diagnostic pathology | Blood donors | Healthy blood donors |
| SAMPLE SIZE | 235 donors | 5,339 | 1,000 | 1,559 |
| SAMPLING METHOD | Random sampling | Random sampling | Random sampling | Random sampling |
| SEROPREVALENCE | 1,7% positives cases | Thirty-eight of 5339 specimens were IgG-positive (general pathology, 19 of 3231; prenatal screening, 7 of 560; plasmapheresis donors, 12 of 1548)The adjusted seroprevalence estimate among those with general pathology blood tests (all ages) was 0.15% and 0.29% and 0.29% for plasmapheresis donors (20–69 years). In people aged 20–39 years, the common age group for all three collection groups, the adjusted seroprevalence estimate was 0.24% for the general pathology group, 0.79% for the prenatal screening group, and 0.69% for the plasmapheresis donors. Overall the seroprevalence is estimated to be less than 1% | 13.7% positivity | With SERA (Serum Epitope Repertoire Analysis), we observed a significant increase in SARS-CoV-2 seropositivity rates over the four-month period, from 0% [95% CI: 0–1.5%] in March to 11.6% [6.0–21.2%] in July |
| STUDY PERIOD | November 2019-March 2020 | April-June 2020 | June-July 2020 | March-July 2020 |
| RISK FACTORS | Age, young people (between 18–30 years old) | | | Age, gender, race |
| SYMPTOMS | Respiratory distress (23 donors), dyspnea (14 donors), other flu-like/respiratory symptoms (18 donors), chest CT image opacity without clinical symptoms (2 donors), cough (1 donor), fever (1 donor). | | | Asymptomatic cases, mild symptoms |
| NUMBER | 125 | 126 | 127 | 128 |

(*Continued*)

**Table 1.** (Continued)

| NUMBER | 1. | 2. | 3. | 4. |
|---|---|---|---|---|
| REFERENCE | Mahallawi WH, Al-Zalabani AH. **The seroprevalence of SARS-CoV-2 IgG antibodies among asymptomatic blood donors in Saudi Arabia**. Saudi J Biol Sci. mars 2021;28(3):1697-701. | Martinez-Acuña N, Avalos-Nolazco D, Rodriguez-Rodriguez D, Martinez-Liu C, Taméz RC, Flores-Arechiga A, et al. **Seroprevalence of anti-SARS-COV-2 antibodies in blood donors from Nuevo Leon state, Mexico, during the beginning of the COVID-19 pandemic** [Internet]. 2020 Nov [cited 2022 Jan 20] p. 2020.11.28.20240325. Available from: https://www.medrxiv.org/content/10.1101/2020.11.28.20240325v1 | Ng D, Goldgof G, Shy B, Levine A, Balcerek J, Bapat SP, et al. **SARS-CoV-2 seroprevalence and neutralizing activity in donor and patient blood from the San Francisco Bay Area.** medRxiv 2020; published online May 27. DOI:10.1101/2020.05.19.20107482 (preprint). | Ojal J, Brand SPC, Were V, Okiro EA, Kombe IK, Mburu C, et al. **Revealing the extent of the COVID-19 pandemic in Kenya based on serological and PCR-test data** [Internet]. medRxiv; 2020 [cited 2022 Feb 9]. p. 2020.09.02.20186817. Available from: https://www.medrxiv.org/content/10.1101/2020.09.02.20186817v1 |
| LOCATION/ COUNTRY | Saudi Arabia | Nuevo Leon state, Mexico | San Francisco Bay Area | Kenya, Africa |
| SELECTION CRITERIA | Men and women, all ages | Men and women, blood donors living in Nuevo Leon state who attended two donation venues. Donors selected according to the requirements of the Mexican Official Norm NOM-253-SSA1-2012 | Men and women, 18 years old and over | Men and women, all ages |
| TYPE OF POPULATIONS | Asymptomatic Blood donors | Blood donors | Blood donors | Residents, blood donors |
| SAMPLE SIZE | 1,212donors | 1,968 | 2,000 | 320,000 |
| SAMPLING METHOD | Random sampling | Based on geographic areas and blood bank center Random sampling | Random sampling | Random sampling |
| SEROPREVALENCE | The seroprevalence of SARS-CoV-2 among blood donors in donors in Al-Madinah was 19.31% (n = 234/1212). (n = 234/1212) | 3.99% | Seropositivity 0.1% in 1,000 blood donors | 5.2%-41% |
| STUDY PERIOD | May-July 2020 | 1. January- 30. August 2020 | March 2020 | February-July 2020 |
| RISK FACTORS | | Age | | |
| SYMPTOMS | Asymptomatic cases | | | |
| NUMBER | 129 | 130 | 131 | 132 |
| REFERENCE | Percivalle E, Cambiè G, Cassaniti I, Nepita EV, Maserati R, Ferrari A, et al. **Prevalence of SARS-CoV-2 specific neutralising antibodies in blood donors from the Lodi Red Zone in Lombardy, Italy, as at 06 April 2020**. Euro Surveill Bull Eur Sur Mal Transm Eur Commun Dis Bull. juin 2020;25(24). | Saeed S, Drews SJ, Pambrun C, Yi Q-L, Osmond L, O'Brien SF. **SARS-CoV-2 seroprevalence among blood donors after the first COVID-19 wave in Canada**. Transfusion (Paris). mars 2021;61 (3):862-72. | Slot E, Hogema BM, Reusken CBEM, Reimerink JH, Molier M, Karregat JHM, et al. **Herd immunity is not a realistic exit strategy during a COVID-19 outbreak** [Internet]. 2022 [cited 2022 Jan 18]. Available from: https://www.researchsquare.com/article/rs-25862/v1 | Sughayer MA, Mansour A, Nuirat AA, Souan L, Ghanem M, Siag M, et al. **Dramatic Rise of Seroprevalence Rates of SARS-CoV-2 Antibodies among Healthy Blood Donors: The evolution of a Pandemic** [Internet]. 2021 Mar [cited 2022 Jan 20] p. 2021.03.02.21252448. Available from: https://www.medrxiv.org/content/10.1101/2021.03.02.21252448v1 |
| LOCATION/ COUNTRY | Lodi Red Zone, Italy (10 municipalities) | Canada | Netherlands | Jordan |
| SELECTION CRITERIA | 272 Men, 118 Women, 19–70 years old, median age 46 years | Men and women, 18 years old and over | Men and women, be completely healthy at the time of donation, 18 years old and over | Healthy asymptomatic subjects between the ages of 18 and 63 who underwent routine screening to determine their acceptability for donation. With asymptomatic cases |

(Continued)

**Table 1.** (Continued)

| NUMBER | 1. | 2. | 3. | 4. |
|---|---|---|---|---|
| TYPE OF POPULATIONS | Registered blood donors, with asymptomatic cases | Blood donors throughout the country | Blood plasma donors and convalescent plasma donors | Healthy blood donors |
| SAMPLE SIZE | 390 | 74,642 donors | 7,361 | 1,374 |
| SAMPLING METHOD | Random sampling | Random sampling | By Age, gender, and zip code of the subject's residence Random sampling | Three batches based on period of sampling |
| SEROPREVALENCE | 20 positive cases of 390 5.13% | Blood donors nationwide 552/74642 donors had detectable antibodies, the adjusted seroprevalence was 7.0/1000 (0.7%) donors. Prevalence was differential by geography Ontario had the highest rate at 8.8/1000 donors, compared to the Atlantic region at 4.5/1000 donors; adjusted odds ratio (aOR) 2.2. Donors who self-identified as an ethnic minority donors were more likely than white donors to be seroreactive a OR 1.5 | 3.1% (230/7361) | January to september 2020: 0% (95% CI 0.00%, 0.51%) (two first groups) Late January and early February 2020: 27.4% (95% CI 22.5% and 32.9%) (third group) |
| STUDY PERIOD | April-2020 | May-July2020 | 1. April-15. April 2020 | January 2020-February 2021 |
| RISK FACTORS | The most densely populated centres were the most affected by COVID-19 | | Age (aged 18–30 years) | |
| SYMPTOMS | Fever, fatigue, cough, cold sore throat, anosmia and dysgeusia, muscular pain, diarrhoea, asymptomatic cases | | Mild symptoms | Asymptomatic cases |
| NUMBER | 133 | 134 | 135 | 136 |
| REFERENCE | Sykes W, Mhlanga L, Swanevelder R, Glatt TN, Grebe E, Coleman C, et al. **Prevalence of anti-SARS-CoV-2 antibodies among blood donors in Northern Cape, KwaZulu-Natal, Eastern Cape, and Free State provinces of South Africa in January 2021**. Res Sq. 12 févr 2021; | Thompson CP, Grayson NE, Paton RS, Bolton JS, Lourenço J, Penman BS, et al**. Detection of neutralising antibodies to SARS-CoV-2 to determine population exposure in Scottish blood donors between March and May 2020**. Eurosurveillance. 2020 Oct 22;25(42):2000685. | Uyoga S, Adetifa IMO, Karanja HK, Nyagwange J, Tuju J, Wanjiku P, et al. **Seroprevalence of anti-SARS-CoV-2 IgG antibodies in Kenyan blood donors** [Internet]. 2020 Jul [cited 2022 Jan 20] p. 2020.07.27.20162693. Available from: https://www.medrxiv.org/content/10.1101/2020.07.27.20162693v1 | Valenti L, Bergna A, Pelusi S, Facciotti F, Lai A, Tarkowski M, et al. **SARS-CoV-2 seroprevalence trends in healthy blood donors during the COVID-19 Milan outbreak** [Internet]. 2020 May [cited 2022 Jan 20] p. 2020.05.11.20098442. Available from: https://www.medrxiv.org/content/10.1101/2020.05.11.20098442v2 |
| LOCATION/COUNTRY | The Northern Cape, provinces, KwaZulu-Natal, Eastern Cape and the Free State, South Africa | Scotland | Kenyan, Africa | Milan, Northern Italy |
| SELECTION CRITERIA | Men and women, 15–69 years old | Men and women, 16 years old and over | Blood donors aged 16–65 years, weighing ≥50kg, with haemoglobin of 12.5g/dl, a normal blood pressure (systolic 120–129 mmHg and diastolic BP of 80–89 mmHg), a pulse rate of 60–100 beats per minute and without any history of illness in the past 6 months | Healthy asymptomatic adults aged 18–70 years old |
| TYPE OF POPULATIONS | Blood donors | Blood donors | Blood donors | Blood donors |
| SAMPLE SIZE | 4,858 donors | 3,500 | 3,098 | 789 |
| SAMPLING METHOD | Random sampling | Based on geaographical areas (region) and period. Random sampling | Random sampling | Random sampling |

(*Continued*)

**Table 1.** (Continued)

| NUMBER | 1. | 2. | 3. | 4. |
|---|---|---|---|---|
| SEROPREVALENCE | EC-63% (2.8%), NC-32% (2.2%), FS-46% (2.4%) and ZN-52% (2.4%) with Eastern Cape (EC), Free State (FS), KwaZulu Natal (ZN) and Northern Cape (NC) | 3,17% (111/3500) | Crude overall seroprevalence: 5.6% (174/3098) Population-weighted, test-adjusted national seroprevalence: 5.2% (95% CI 3.7–7.1%) | Overall adjusted: 2.7%, 95% c.i. 0.3–6% |
| STUDY PERIOD | January-2020 | March-May 2020 | 30. April- 16. June 2020 | 24. February-8. April 2020 |
| RISK FACTORS | | Geographical areas, health board | Region, age | Age, rate of triglycerides, eosinophils, and lymphocytes |
| SYMPTOMS | | | | Asymptomatic cases |
| NUMBER | 137 | 138 | 139 | |
| REFERENCE | Vassallo RR, Bravo MD, Dumont LJ, Hazegh K, Kamel H. **Seroprevalence of Antibodies to SARS-CoV-2 in US Blood Donors** [Internet]. 2020 Sep [cited 2022 Jan 20] p. 2020.09.17.20195131. Available from: https://www.medrxiv.org/content/10.1101/2020.09.17.20195131v1 | Villarreal A, Rangel G, Zhang X, Wong D, Britton G, Fernandez PL, et al. **Performance of a Point of Care Test for Detecting IgM and IgG Antibodies Against SARS-CoV-2 and Seroprevalence in Blood Donors and Health Care Workers in Panama**. Front Med. 2 mars 2021;8:616106. | Younas A, Waheed S, Khawaja S, Imam M, Borhany M, Shamsi T. **Seroprevalence of SARS-CoV-2 antibodies among healthy blood donors in Karachi, Pakistan**. Transfus Apher Sci Off J World Apher Assoc Off J Eur Soc Haemapheresis. déc 2020;59 (6):102923. | |
| LOCATION/ COUNTRY | United States | Panama | Karachi, Pakistan | |
| SELECTION CRITERIA | Men and women, 18–64 years old | Men and women, 18 years old and over | Only men, 30–37 years old | |
| TYPE OF POPULATIONS | Blood donors | Blood donors and health workers (healthy volunteer) | Healthy blood donors who are venaked to the National Institute for Bloodand Bone Marrow Transplantation between May June and July 2020. But also those who came in October 2019 | |
| SAMPLE SIZE | 252,882 | 702 | 380 donors | |
| SAMPLING METHOD | Random sampling | Random sampling | Random sampling | |
| SEROPREVALENCE | Unique donors (n = 252,882) showed an overall seroprevalence in June (1.37%) and July (2.26%), with the highest prevalence in northern New Jersey (7.3%) | 97.2% (95%CI 84.2–100.0%) to detect both IgM and IgG. The analysis showed a Kappa of 0.898 (95%CI 0.811–0.985) and 0.918 (95%CI 0.839–0.997) for IgM and IgG, respectively We found an overall antibody seroprevalence of 11.6% (95% CI 8.5–15.8%) health care workers and healthy blood donors | 40% | |
| STUDY PERIOD | 1. June-31. July 2020 | April-July 2020 | June-2020 | |
| RISK FACTORS | Gender, age, race/ethnicity, education level, collection site types, location | | | |
| SYMPTOMS | | | Mild symptoms | |

## Serological status

The determination of seroprevalence in most studies involved the ELISA test, which detects IgG and IgM antibodies. Other studies used a serum epitope repertoire analysis or a plaque

reduction neutralization test, and IgA antibodies were also identified. Estimated seroprevalence was 0 to 69% (Table 1). Among the 133 original studies, 13 studies investigated IgG [11–23], 8 studies IgG or IgM [24–31], two studies IgG and IgM [32,33], one study IgA or IgG [34], one study IgG only or IgG and IgM [35] and one study IgG or IgM or both simultaneously [36].

## Prevalence by continent and over time

**General population.**    The most-represented and explored territories were Europe, with 34 studies, distributed among Italy (9), France (4), England (4), Spain (3), Germany (3), Slovenia (2), Switzerland (2), Luxembourg (1), Austria (1), Hungary (1), Denmark (1), Faroe Island (1), Greece (1), Albania (1), Sweden (1), United Kingdom (1), Georgia (1), Poland (1), and Estonia (1); the United States, with 24 studies; other Organisation for Economic Co-operation and Development (OECD) countries, with Japan (3), Canada (1), and Australia (1); and other countries, with India (12), Brazil (6), China (6), Iran (4), Argentina (1), Iraq (1), Palestine (1), Pakistan (1), Qatar (1). South Africa (1), and Sudan (1). Fig 2 represents the spread of seroprevalence estimates in each continent over the study period.

Most studies (n = 52) estimated a seroprevalence of SARS-CoV-2 of 0 to 7%; for more than half, the seroprevalence was < 5%. England had the highest seroprevalence, estimated at 69% from July to September 2020, followed by Iraq, at 62.6% between July 2020 and February 2021.

**Blood donors.**    The most represented and explored territories were Europe, with 10 studies, distributed among Italy (3), France (2), Germany (1), Denmark (1), Netherlands (1), Sweden (1), and Scotland (1); the United States, with 7 studies; other OECD countries, with Canada (1), Australia (1) and Mexico (1); and other countries, with Kenya (3), Brazil (2), Saudi Arabia (2), China (1), South Africa (1), Pakistan (1), and Panama (1).

Most studies (n = 20) estimated a seroprevalence of SARS-CoV-2 of 0 to 7%; for more than half, the seroprevalence was < 5% (n = 17).

South Africa had the highest seroprevalence, estimated at 63% in January 2021, and the lowest seroprevalence was estimated in the United States, 0% in March 2020.

**Symptoms.**    A total of 84 articles [3,4,13,15,16,18,21,23,29,31,32,35,37–108] focused on COVID-19–related symptoms. The seroprevalence of asymptomatic cases ranged from 0%

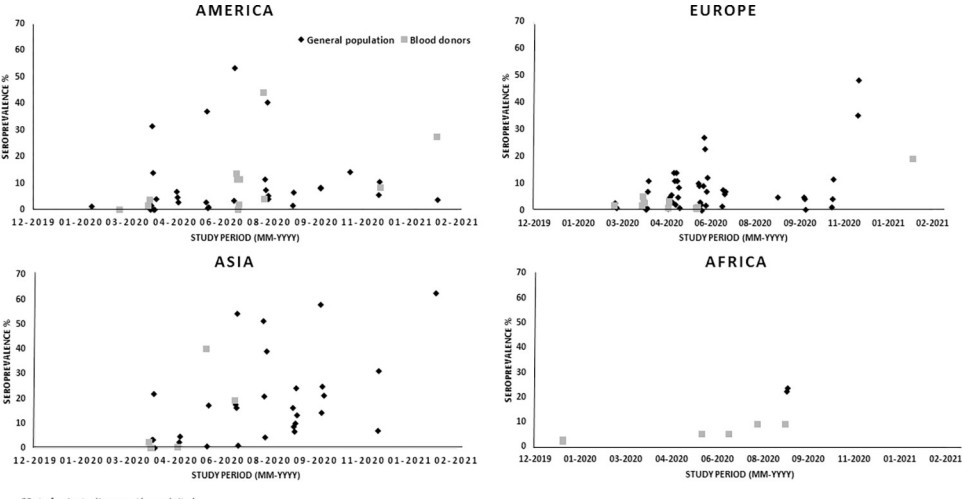

**Fig 2. Seroprevalence (%) in each continent over the observation period.**

(Q1-Q3 1–3%) to 31.5% (3–7%). Only 35 articles explicitly assessed asymptomatic populations [18,19,24–26,32,45,47,53,56,59,64,66,69,75,76,78,81,83,89,90,92,93,95,98,99,102,105–114]. The symptoms most strongly associated with seropositivity were anosmia, agueusia, fever, fatigue, rhinorrhea, sore throat, breathing difficulties, flu-like symptoms, cough, dyspnoea, myalgias, headache and asthenia. Diarrhoea, muscle aches, and chest pain were also common.

**Risk factors.** The most densely populated centres were the most affected by COVID-19 [8,16,20,32,40,41,48,50,53,55,56,59,71,76,78,81,86,87,88,91,92,94,95,101,102,106,115–126]. Sex was examined in 14 studies, which found mostly no significant difference (one a higher seroprevalence in men and 6 a higher seroprevalence in women).

In articles published by the end of 2020 [91,95,99,118], the age group most significantly affected was young adults aged 18–30 years. Other studies showed an increase in seroprevalence in older age groups. For children, eighteen studies [37,47,49,56,60,68,75,79,83,85,88,89,93,108,119,122,127,128] showed significantly lower seroprevalence than for adults. For articles published after late 2020, the age ranges were narrower than in articles published at the beginning of the pandemic. Since the emergence of the virus, young populations have also been found to have high seroprevalence and should not be overlooked [40,96]. The impact of social disadvantage was documented in the general population and blood donors in a comparison of seroprevalence according to income [32,47,68,104,126,129–134] or education level [18,19,20,47,48,53,57,68,82,99,130–138]. Prevalence was higher in the lowest than highest income groups (prevalence doubled [32], from 1.8% to 3.7%). The seroprevalence was two times higher with low than high education [91]. Thirteen studies showed that the risk of seropositivity increased by about 30% with a confirmed COVID-19 case in the household [23,32,36,47,55,59,68,75,79,88,92,125,136]. Ten studies showed that the risk of seropositivity increased with the number of children in the household [16,25,55,71,77,88,116,120,125,133]. The highest seroprevalence was found in the most deprived areas [59,67,68,86,92,104,126,130–133,139,140]. Finally, seven studies [18,60,68,87,93,116,141] found a decrease in SARS-CoV-2 seroprevalence associated with greater freqency of smoking.

## Discussion

Published studies of seroprevalence of SARS-CoV-2 in the general population over the year after the onset of the pandemic up to early April 2021 showed estimates ranging from 0% in Palestine in the West Bank between June and July 2020 to 69% in England between July and September 2020, with different dynamics across continents.

This review purposely focused on the first year of the pandemic to better understand the spread and dynamics from the early stages of the pandemic and to summarize identified factors for SARS-CoV-2 penetration that could further serve as a reference for adapted measures to mitigate future epidemics. Such measures include barriers, social distancing, evolving vaccines according to the molecular and biological monitoring of viruses, and preventive or early treatments to avoid severity.

Fig 2 shows that the blood donors were not fully representative of the general population. Their seroprevalence and respective peaks show variations of lower magnitude in general. Blood donors could have been an early resource for documenting the epidemic before their decrease in frequency, with population-based studies taking place later. Some continents have fewer donors than others, probably related to national heath care organisation.

The variations observed across studies, besides the true virus exposure and spread differences over space and time within countries, could be due to different sampling techniques and

the use of different serological tests. Our results suggest that the number of symptomatic cases was lower than the number of actual cases, despite only few studies (n = 21) characterizing asymptomatic seropositive cases.

Most of the studies did not find or found minimal differences regarding sex. Age categories analysis did not yet reveal any conclusive results. Social disadvantage seemed to play a role, at least for the least well-off categories, but the impact of belonging to the most privileged categories, whatever the classification used, remains to be elucidated. Finally, the presence of a COVID-19 case in a household increasing the risk of the other members has been demonstrated consistently for developing antibodies.

This review had a longer study period than six previous reviews; it focused on general populations and blood donors; it covered a larger world area by including all continents as compared with Grant et al. [8], Rostami et al. [10], Chen et al. [6] and Levesque and Maybury[9], and it summarised a number of risk factors identified, mostly of a sociodemographic nature.

A limitation of this scoping review is the heterogeneity of samples with different age ranges, so seroprevalence data are not fully comparable. Potential biases in such seroprevalence observational studies also do not account for SARS-COV-2–related deaths. Also, we did not include grey literature because we did not know how to search such literature in the particular context of this pandemic with so much suspicion of non-peer–reviewed publications. A third limitation is that we did not assess the methodological quality of the studies reviewed. Finally, the results of seroprevalence studies may have been affected by the specificity and sensitivity of different serological methods used.

In conclusion, from this scoping review of seroprevalence studies over the first year of the COVID-19 pandemic, the seroprevalence of SARS-CoV-2 varied according to the study period, with lower seroprevalence at the beginning of the epidemic than between July and September 2020. This review documented the progression of this virus across the world in time and space and the risk factors that influenced its spread.

## Supporting information

**S1 Checklist. PRISMA-ScR.**
(DOCX)

## Author Contributions

**Conceptualization:** Hélène Jeulin, Anne Gegout-Petit, Evelyne Schvoerer, Francis Guillemin.

**Data curation:** Clémentine Metzger, Taylor Leroy, Agathe Bochnakian.

**Resources:** Taylor Leroy, Francis Guillemin.

**Supervision:** Francis Guillemin.

**Writing – original draft:** Clémentine Metzger.

**Writing – review & editing:** Taylor Leroy, Agathe Bochnakian, Hélène Jeulin, Anne Gegout-Petit, Karine Legrand, Evelyne Schvoerer, Francis Guillemin.

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
