## [Decision Letter · Decision Letter 0]

6 Dec 2021

PONE-D-21-32219Seroprevalence of SARSCoV-2
in general populations: a scoping review over one year
pandemic.PLOS ONE

Dear Dr. METZGER,

Thank you for submitting your manuscript to PLOS ONE. After careful consideration, we
feel that it has merit but does not fully meet PLOS ONE’s publication criteria as it
currently stands. Therefore, we invite you to submit a revised version of the
manuscript that addresses the points raised during the review
process. Please submit
your revised manuscript by Jan 20 2022 11:59PM. If you will need more time than this
to complete your revisions, please reply to this message or contact the journal
office at plosone@plos.org. When you're ready to
submit your revision, log on to https://www.editorialmanager.com/pone/ and select the 'Submissions
Needing Revision' folder to locate your manuscript file.

Please include the following items when submitting your revised
manuscript:A rebuttal letter that responds to each point raised by the academic
editor and reviewer(s). You should upload this letter as a separate file
labeled 'Response to Reviewers'.A marked-up copy of your manuscript that highlights changes made to the
original version. You should upload this as a separate file labeled
'Revised Manuscript with Track Changes'.An unmarked version of your revised paper without tracked changes. You
should upload this as a separate file labeled 'Manuscript'.

If you would like to make changes to your financial disclosure, please include your
updated statement in your cover letter. Guidelines for resubmitting your figure
files are available below the reviewer comments at the end of this letter.

We look forward to receiving your revised manuscript.

Kind regards,

Chandrabose Selvaraj, Ph.D.

Academic Editor

PLOS ONE

Journal Requirements:

[This research was funded by METROPOLE DU GRAND NANCY]

 [Funding : The study was funded by Metropole du Grand (https://www.grandnancy.eu/accueil/).

The funder had no role in study design, data collection and analysis, decision to
publish, or preparation of the manuscript.]

Reviewers' comments:

Reviewer's Responses to Questions

5. Review Comments to the Author

Reviewer #1: This was a review of SARS-CoV-2 seroprevalence between January 2020 and
April 2021 that included 68 studies. While I applaud the authors for their efforts
to narratively synthesis a heterogenous group of literature, and recognize the
challenges of conducting evidence synthesis during the high-volume publication
period of COVID-19, I have the following major concerns:

1) Given that this is a scoping review, the authors may need to review and adhere to
the PRISMA scoping review guidelines. This template will help the authors to ensure
the of their methods and reporting.

2) Scope and search. This was called a scoping review however, I'm not sure that it
meets the methodological requirements of a scoping review. There were only 2
databases searched and the search strategy was quite rudimentary, with only a few
terms. Scoping reviews are intended to cover as much of the available literature as
possible, which typically involves searching multiple databases and the grey
literature using a robust search strategy with involvement of a librarian. However,
this search does not accomplish that. Was grey literature included? If not, why not?
Was a health sciences librarian consulted about the search strategy? If not, why
not? For this to be called a scoping review the search databases and search strategy
need to be expanded and made more robust.

3) Inclusion/exclusion criteria. Please could the authors define a key aspect of the
inclusion criteria: what is a general population?

4) Missing original studies and reviews. This is a fairly substantial issue with this
article. The authors indicated that they included original studies and reviews.
However, they are missing hundreds of original studies and some of the landmark
systematic reviews on SARS-CoV-2 seroprevalence. This is demonstrated by an
examination of the landmark reviews, which included many more original studies,
despite being conducted at earlier points in the pandemic. All of these studies were
available in peer-review or pre-print form prior to the end of the authors search
date:

A) Bobrovitz PLOS One (n=590 studies in the general population; 16(6): e0252617)
(preprint available prior to end of authors search date)

B) Chen in Lancet Global Health (n=84 studies in the general population; Volume 9,
ISSUE 5, e598-e609, May 01, 2021)

C) Rotami Clin Microbiol Infect (n=68 studies in the general population; 2021
Mar;27(3):331-340)

These landmark studies were also not discussed in the discussion section. Can the
authors please explain why these reviews were not included or emphasized? And why
the discrepancies in the number of original studies? To be a scoping review, you
would expect that these studies would be included.

5) Search date. Unfortunately, the literature on SARS-Cov-2 moves very quickly. I
know it is very challenging to keep up to date with evidence synthesis and
appreciate that limited resources and time play a role. However, this manuscript is
already outdated. I know this is a difficult review comment to receive when doing
evidence synthesis and for that I am sorry. But, according to the serotracker.com
online database of SARS-CoV-2 - several hundred studies in the general population
have been published. Given the significant literature published after the end of the
search date, the authors would need to update their search to include some of these
studies. Or at the very least explain how their snapshot from January 2020 - April
2021 provides added value in the context of the other larger and more inclusive
reviews that have already been published.

6) Results. The authors do not do draw a distinction when reporting results of
reviews and original studies. These are different types of evidence and the
reporting of their results should reflect this.

Reviewer #2: This scoping review summarizes SARS-CoV-2 seroprevalence studies mainly
covering the first/second pandemic waves internationally. This study is of high
interest and could help to gain rapid and broad insights related to spread/dynamics
from the early stages of the pandemic. However, there are some limitations and
important (major) revisions need to be done. Some parts (see below) are kept very
generic and are not structured well. While the study aims to summarize the
seroprevalence progression over time and according to geographic locations, this is
not clearly shown in the article (in general the specific aims and the mapping
approach is not very clear). There is no clear strategy for the qualitative data
synthesis and results should be written more clearly. Thus, it is hard to draw
conclusion (e.g. it is written that the increase is exponential in the abstract, but
this is not clearly reflected in the results. It’s generally not surprising that the
prevalence increased over time. In my option it would be much more interesting to
qualitatively describe the changes/locations in some more detail which has important
implications.

Abstract: there is some information missing. Please provide some further information
regarding eligibility (e.g all ages? Children as well?). The method is not stated.
Was the data only qualitatively summarized, if so, how? Suggest not to abbreviate
Web of Science. A conclusion is missing.

Introduction:

The aims are very broad. It should be stated that the study aims to summarize/map the
studies according to time elapsed since pandemic outbreak, geographic region,
summarizing risk factors etc.

Method:

in general, the structure of the methods section should be improved. Some aspects
need clarification. For example, the authors start with a statement regarding
databases that were searched and state further below that also preprints were
searched (..to retrieve further articles from reference list screening). I suggest
providing subtitles that contain eligibility, search strategy, data extraction.

Please clarify eligibility criteria: which languages? Age range? Design of studies
etc.

Search strategy is unclear. Was the Search done only with key terms? Or MESH terms
such as "Seroepidemiologic Studies"[Mesh]. The search strategy is not very broad –
there is potential for missing studies.

The selection process is not clearly stated. How was the stage 1 (title, abstract)
and stage 2 done? Independently by two reviewers?

Data extraction. The variables extracted are not well defined…what is meant by
“…calibrated form..” “…were seropravalence…”. Was this done by one or two authors?
Why are the variables displayed in the tables not described in this part?

Results:

In general: please provide a clear structure with subtitles if necessary (Study
characteristics, prevalence by continent and over time, symptoms, risk factors,
etc.

Figure 1: some inconsistencies with spacing’s before, after etc. I wonder why the
duplicates were not removed earlier in the process, its written that removal was
done after full-text analysis? The preprint articles from the reference list
screening do not appear in the flow-diagram.

Line 100: do you mean with OCDE? Is Iran an OECD country? I suggest order this part
according to the geographical location (as in the Figure 2).

Figure 2: please clearly indicate what is shown in this figure. Please add some
description in the main text as well as in the Figure caption. It should be clear
what comparison is shown here. Is it the lowest and the highest prevalence by
continent? I think it would be much more helpful to show the % separated (all
prevalence estimates) by continent and ordered by time. This would allow good
comparisons.

Supplementary table: The items displayed do not match the variables planned to be
extracted in the methods section. Please use the term sample instead of population
(which is usually the target for the estimates). I strongly suggest to include the
table in the manuscript rather than in the supplement. In any case, the table layout
should be improved (format as references, repeat the titles for each page, exact
dates or at least clearly declare time points of assessments). The summary for
seroprevalence should be consistent and self-explanatory. E.g. 9 of 904 instead of
percent, first round, second round (what does it mean?)

Line 113: I do not understand this sentence. What is Q1-Q3? Quartiles for the
minimum?

Line 125: I suggest moving this to the study characteristics section

Discussion

Line 150: I do not understand this sentence

Line 155: this should be stated more clearly already in the results in order to make
this statement here.

Para 2: I think statements such as “Populations were not differently affected….” are
not allowed based on the design of your study. Even if the majority of studies found
no difference, this statement is not correct (if a meta-regression would have been
conducted, some statements in this direction could have been made). Statements like
most studies…found no…would be ok.

---

## [Author Response · Author response to Decision Letter 0]

26 Feb 2022

Journal Requirements:

Requirement 1. Please ensure that your manuscript meets PLOS ONE's style
requirements, including those for file naming. The PLOS ONE style templates can be
found at 

'author response' We have checked title and manuscript format requirements

'author action' We have applied those requirements

Requirement 2. Thank you for stating the following in the Acknowledgments Section of
your manuscript: 

[This research was funded by METROPOLE DU GRAND NANCY]

 [Funding : The study was funded by Metropole du Grand (https://www.grandnancy.eu/accueil/).

The funder had no role in study design, data collection and analysis, decision to
publish, or preparation of the manuscript.]

'author response' We updated and put the funding information in the Funding Statement
section and removed any funding information from other places.

'author action' The Funding Statement section now reads: “This research was funded by
Metropole du Grand Nancy, CHRU de Nancy, and Inserm. The funders had no role in
study design, data collection and analysis, decision to publish, or preparation of
the manuscript.”

Requirement 3. We note that you have stated that you will provide repository
information for your data at acceptance. Should your manuscript be accepted for
publication, we will hold it until you provide the relevant accession numbers or
DOIs necessary to access your data. If you wish to make changes to your Data
Availability statement, please describe these changes in your cover letter and we
will update your Data Availability statement to reflect the information you
provide.

'author response' We did not mean to provide repository information for our data.We
misunderstood the instructions to authors. Actually, all data are in the manuscript
and supplementary files, and the data applicability statement is not applicable. 

'author action' Our Data availability statement has been changed to: “Not
applicable”

Requirement 4. Please include captions for your Supporting Information files at the
end of your manuscript, and update any in-text citations to match accordingly.
Please see our Supporting Information guidelines for more information: http://journals.plos.org/plosone/s/supporting-information. 

'author response' According to Reviewer #2’s suggestion, we have included the table
into the manuscript instead of the supplement. The in-text citation and the
supporting information file have been updated accordingly, and captions are provided
at the end of the manuscript.

'author action' The following captions are added at the end of the manuscript : 

Figure 1: PRISMA Flowchart

Figure 2: Seroprevalence (%) in each continent over the observation period.

Table 1. Characteristics of country, methods, population of seroprevalence of the
study.

'additional author action' To better reflect the objective of this article, the title
has been slightly amended into: 

“Seroprevalence and SARS-CoV-2 invasion in general populations: a scoping review over
the first year of the pandemic.”

Considering the important contribution of one author (TL), all other authors agreed
to modify authors ranking.

 Reviewer #1

'reviewer comment' This was a review of SARS-CoV-2 seroprevalence between January
2020 and April 2021 that included 68 studies. While I applaud the authors for their
efforts to narratively synthesis a heterogenous group of literature, and recognize
the challenges of conducting evidence synthesis during the high-volume publication
period of COVID-19, I have the following major concerns:

1) Given that this is a scoping review, the authors may need to review and adhere to
the PRISMA scoping review guidelines. This template will help the authors to ensure
the of their methods and reporting.

'author response' We thank the reviewer for her/his appreciation and the advice.
Actually, we used the PRISMA-Scr guidelines / checklist at first submission and this
was submitted with the manuscript in the supporting information file. 

'author action' After revision, we have updated our PRISMA-Scr checklist, and
attached it in the revised Supporting information file.

'reviewer comment' 2) Scope and search. This was called a scoping review however, I'm
not sure that it meets the methodological requirements of a scoping review. There
were only 2 databases searched and the search strategy was quite rudimentary, with
only a few terms. Scoping reviews are intended to cover as much of the available
literature as possible, which typically involves searching multiple databases and
the grey literature using a robust search strategy with involvement of a librarian.
However, this search does not accomplish that. Was grey literature included? If not,
why not? Was a health sciences librarian consulted about the search strategy? If
not, why not? For this to be called a scoping review the search databases and search
strategy need to be expanded and made more robust.

'author response' We searched 2 databases for the whole period and searched medRXiv
for the early months of 2021 only, on the premise that older paper would have been
published, and papers in this database would have not been peer-reviewed. We realize
this was insufficient. Moreover this was mentioned in a wrong place in our method
section. 

We have now extended the medRXiv search to the whole period, i.e. cumulating 3
databases, mentioned it the method section and flow chart, and integrated the
results. 

The key words for the search strategy was the final retained after several variations
that did not bring more articles to our scrutiny. 

We did not include grey literature search, for we could not know how to search such
literature in the particular context of this pandemic where so much suspicion was
brought up for those non peer-reviewed publications.

We could not afford, neither our institution, to involve a health sciences librarian.
We managed to determine the search strategy collectively with all co-authors. 

To see whether we could make our search strategy more robust, we applied search
strategies of other reviews (those we previously identified and those mentioned by
this reviewer below) on our review period. As can be seen in the table below, the
number of identified studies are largely diverse and though do not correlate with
the number of studies retained by each author. Also, we can point that the reviews
conducted by Chen, et al. or Bobrovitz, et al. did not target the same purpose than
our (‘summarise serological surveys for SARS-COV-2 infections in humans’, and
‘seroprevalence surveys published in 2020’, respectively) i.e. all type of
populations. By comparing more closely with Grant, et al. and with Rostami, et al.,
whose purpose were closer to our (‘population-based seroprevalence studies from
Europe available as of 15 september 2020’ and ‘estimate the global and regional
seroprevalence of SARS-COV-2 in people of the general population‘, respectively), we
certainly missed some studies, namely 39 out of which 13 were relevant to our
search, while we identified 9 relevant studies that they did not.

Therefore, we reinforced our search strategies in two ways: 1) by looking more
systematically at previous reviews, including their references, and 2) by searching
medRXiv for the whole period, to incorporate relevant individual studies from these
two sources.

Table 1 : Search strategy from each author applied to our search period (period 1:
Jan 1st, 2020 – April 10, 2021) and to more recent period (period 2: april 11th,
2021 – december 31st, 2021) 

Pubmed medRxiv Web of science

Bobrovitz

Period 1 : N=136 780

Period 2 : N=53 313 Period 1 : N=7 057

Period 2 : N=6 219

Chen

Period 1 : N=8 478

Period 2 : N=6 762 Period 1 : N=3 924

Period 2 : N=2 703 Period 1 : N=5 912

Period 2 : N=5 773

Grant

Period 1 : N=659

Period 2 : N=601 Period 1 : N=591

Period 2 : N=384 

Metzger

Period 1 : N=742

Period 2 : N=823 Period 1 : N=586

Period 2 : N=709 Period 1 : N=281

Period 2 : N=521

Rostami

Period 1 : N=11 126 (10 701+425)

Period 2 : N=7 756 (7 480+276) 

'author action' The flow chart and method section have been amended accordingly. The
reviews source have been identified. The 71 additional articles, including 6
reviews, were added to the search, analysed and results have been completed and
amended accordingly.

'reviewer comment' 3) Inclusion/exclusion criteria. Please could the authors define a
key aspect of the inclusion criteria: what is a general population?

'author response' This is an important question that may make a difference with other
reviews, also explaining some heterogeneity in the articles selected. We have
considered as general population a population-based sample, or a described general
population sample, preferably (but not exclusively) obtained by random selection
from a large population (survey or database). We have also included blood donors
sample as proxy for general population, since they are likely not biased toward
SARS-COV-2 seroprevalence, and we have presented them distinctly in the results.

Exclusion criteria were health care workers, people attending a clinic or a hospital,
and samples from a professional branch (industry, factory, farmers, university) and
from a particular population (students, nursing home).

To document this search we hand-searched all elicited articles instead of using the
term “general population” in the key words, which proved to be too restrictive, as
can be seen in the table2 below compared to table 1 above

Table 2 : Search strategy from each author applied to our search period (Jan 1st,
2020 – April 10, 2021 ) adding « general population » in key-words

Pubmed medRxiv Web of science

Bobrovitz

N=6 987

 N=288

Chen

N=440

 N=3 425

 N=173

Grant

N=69

 N=530

Metzger

N=131

 N=450

 N=67

Rostami

N=616 (591 + 25)

'author action' This is now made more explicit in the Method section.

'reviewer comment' 4) Missing original studies and reviews. This is a fairly
substantial issue with this article. The authors indicated that they included
original studies and reviews. However, they are missing hundreds of original studies
and some of the landmark systematic reviews on SARS-CoV-2 seroprevalence. This is
demonstrated by an examination of the landmark reviews, which included many more
original studies, despite being conducted at earlier points in the pandemic. All of
these studies were available in peer-review or pre-print form prior to the end of
the authors search date:

A) Bobrovitz PLOS One (n=590 studies in the general population; 16(6): e0252617)
(preprint available prior to end of authors search date)

B) Chen in Lancet Global Health (n=84 studies in the general population; Volume 9,
ISSUE 5, e598-e609, May 01, 2021)

C) Rotami Clin Microbiol Infect (n=68 studies in the general population; 2021
Mar;27(3):331-340)

These landmark studies were also not discussed in the discussion section. Can the
authors please explain why these reviews were not included or emphasized? And why
the discrepancies in the number of original studies? To be a scoping review, you
would expect that these studies would be included.

'author response' We agree. This difference was mainly due to our definition of a
general population. 

We initially selected the Chen, et al. and the Grant, et al. reviews, but not the one
by Bobrovitz, et al. nor the Rostami, et al. Those numbers of studies are not always
those retained for their general population definition and would not be for our, as
explained above. 

We have now included 6 reviews in our search and analysis.

'author action' The change in our search strategy has resulted in a large increase of
the studies included (n=133) and the reviews (n=6) that helped identify additional
studies.

'reviewer comment' 5) Search date. Unfortunately, the literature on SARS-Cov-2 moves
very quickly. I know it is very challenging to keep up to date with evidence
synthesis and appreciate that limited resources and time play a role. However, this
manuscript is already outdated. I know this is a difficult review comment to receive
when doing evidence synthesis and for that I am sorry. But, according to the
serotracker.com online database of SARS-CoV-2 - several hundred studies in the
general population have been published. Given the significant literature published
after the end of the search date, the authors would need to update their search to
include some of these studies. Or at the very least explain how their snapshot from
January 2020 - April 2021 provides added value in the context of the other larger
and more inclusive reviews that have already been published.

'author response' This point is important. It can be frequently the case when time
passes with submission to previous journals that, moreover, did not take the effort
to provide review of our manuscript. We appreciate the diplomatic way it is
expressed, and the constructive review and criticism. Unfortunately, launching a new
run of retrieving articles among hundreds of studies and extracting data seems a
harder work that what we can do now, and unfortunately somewhat of an endless
effort. As can be seen in table 1 above, the numbers of article to screen from the
new period april to December 2021 would be about the same as what we already did,
which is beyond our capacity. 

More importantly, the scope of this review was purposedly focused on the first year
of the pandemic to better understand the dynamic and factors for SARS-COV-2
penetration that could serve further as reference for setting up adapted measures to
mitigate future epidemic (barriers, social distancing, evolutive vaccines according
to molecular and biological monitoring of viruses, preventive or early treatments to
avoid severity). 

Besides, we have considerably enriched our search strategy and hence increased the
number of relevant studies included (see 2) scope and search above). Our review
points on the following: 

- showing how the pandemic increasing trend of seroprevalence has developed in time
and space over the firs year,

- and documenting associated factors, in particular social factors exposing
individuals to SARS-COV-2 infection.

'author action' The scope and interest of focusing on the first year of the pandemic
are better specified in the objective and the discussion, supported by a more
comprehensive review of studies published over the period.

'reviewer comment' 6) Results. The authors do not do draw a distinction when
reporting results of reviews and original studies. These are different types of
evidence and the reporting of their results should reflect this.

'author response' Thank you for this advice. We have considered presenting the
results separately. Actually the scope of this review was not to produce a synthesis
of seroprevalence estimates or a meta analysis of risk factors for infection, but to
identify and present the heterogeneity of estimates over time and space. Therefore
we used the reviews and meta-analysis to identify missing studies, and we have
presented the new full set of original studies only. However, we discuss the
differences with previous reviews in the discussion.

'author action' Results of all original studies are presented. Previous reviews are
used to identify missing studies and discussed in the discussion. 

 Reviewer #2

 Line numbers refer to those in the original manuscript submission

'reviewer comment' This scoping review summarizes SARS-CoV-2 seroprevalence studies
mainly covering the first/second pandemic waves internationally. This study is of
high interest and could help to gain rapid and broad insights related to
spread/dynamics from the early stages of the pandemic. However, there are some
limitations and important (major) revisions need to be done. Some parts (see below)
are kept very generic and are not structured well. While the study aims to summarize
the seroprevalence progression over time and according to geographic locations, this
is not clearly shown in the article (in general the specific aims and the mapping
approach is not very clear). There is no clear strategy for the qualitative data
synthesis and results should be written more clearly. Thus, it is hard to draw
conclusion (e.g. it is written that the increase is exponential in the abstract, but
this is not clearly reflected in the results. It’s generally not surprising that the
prevalence increased over time. In my option it would be much more interesting to
qualitatively describe the changes/locations in some more detail which has important
implications.

'author response' Thank you for the appreciation, these comments and helpful
suggestions. 

'author action' We have reorganized the method section with more details and have
tried to go more in depth in the qualitative description of the findings.

'reviewer comment' Abstract: there is some information missing. Please provide some
further information regarding eligibility (e.g all ages? Children as well?). The
method is not stated. Was the data only qualitatively summarized, if so, how?
Suggest not to abbreviate Web of Science. A conclusion is missing.

'author response' We acknowledge the abstract was not structured and detailed
enough.

'author action' The abstract has been completed with requested informations

'reviewer comment' Introduction:

The aims are very broad. It should be stated that the study aims to summarize/map the
studies according to time elapsed since pandemic outbreak, geographic region,
summarizing risk factors etc.

'author response' Thank you for your advice to focus more precisely on the review
content. 

'author action' The aim of the study has been rewritten to better cover the work
done.

'reviewer comment' Method:

in general, the structure of the methods section should be improved. Some aspects
need clarification. For example, the authors start with a statement regarding
databases that were searched and state further below that also preprints were
searched (..to retrieve further articles from reference list screening). I suggest
providing subtitles that contain eligibility, search strategy, data extraction.

'author response' Thank you for the valuable advice

'author action' The structure of the method section has been revisited according to
suggestion and to PRISMA-Scr checklist.

'reviewer comment' Please clarify eligibility criteria: which languages? Age range?
Design of studies etc.

'author response' These criteria are now clarified

'author action' The method section has been completed accordingly

'reviewer comment' Search strategy is unclear. Was the Search done only with key
terms? Or MESH terms such as "Seroepidemiologic Studies"[Mesh]. The search strategy
is not very broad – there is potential for missing studies.

'author response' The search strategy was the final retained after several variations
that did not bring more articles to our scrutiny. The use of MESH terms brought much
less studies than key terms. 

'author action' It is now presented in a more clear fashion

'reviewer comment' The selection process is not clearly stated. How was the stage 1
(title, abstract) and stage 2 done? Independently by two reviewers?

'author response' This was done independently by two reviewers, then by a third one
allowing discussion in case of discrepancy after abstract reading to obtain
consensus for selection

'author action' This is now clarified in the method section.

'reviewer comment' Data extraction. The variables extracted are not well defined…what
is meant by “…calibrated form..” “…were seropravalence…”. Was this done by one or
two authors? Why are the variables displayed in the tables not described in this
part?

'author response' Data extraction was conducted by three authors using a standardized
form. 

'author action' This step is now clarified in the method section. Risk factors and
symptoms have been added in the table.

'reviewer comment' Results:

In general: please provide a clear structure with subtitles if necessary (Study
characteristics, prevalence by continent and over time, symptoms, risk factors,
etc.

'author response' Thank you for the advice for clarification

'author action' The text has been amended accordingly.

'reviewer comment' Figure 1: some inconsistencies with spacing’s before, after etc. I
wonder why the duplicates were not removed earlier in the process, its written that
removal was done after full-text analysis? The preprint articles from the reference
list screening do not appear in the flow-diagram.

'author response' Thank you for these relevant remarks. Duplicates were identified
before and sometimes after retrieval when the title in medRxiv and published format
in other databases differed. In the PRISMA flow-chart, full text reading comes
afterwards.

'author action' The figure has been fully revisited and updated

'reviewer comment' Line 100: do you mean with OCDE? Is Iran an OECD country? I
suggest order this part according to the geographical location (as in the Figure
2).

'author response' Done

'author action' We corrected the mistyping and ordered the presentation by
location.

'reviewer comment' Figure 2: please clearly indicate what is shown in this figure.
Please add some description in the main text as well as in the Figure caption. It
should be clear what comparison is shown here. Is it the lowest and the highest
prevalence by continent? I think it would be much more helpful to show the %
separated (all prevalence estimates) by continent and ordered by time. This would
allow good comparisons.

'author response' According to the new set of studies and increase of data collected,
as well as to these remarks, the Figure 2 has revisited and presented in more clear
strategy by time and location.

This is helpful for interpretation of the spread of the pandemic phenomenon.

'author action' Figure 1 is now presenting point estimates over time in each
continent (except Australia where only one estimate was available).

'reviewer comment' Supplementary table: The items displayed do not match the
variables planned to be extracted in the methods section. Please use the term sample
instead of population (which is usually the target for the estimates). I strongly
suggest to include the table in the manuscript rather than in the supplement. In any
case, the table layout should be improved (format as references, repeat the titles
for each page, exact dates or at least clearly declare time points of
assessments).

'author response' In the previous format, time points of assessments were displayed
in an extreme right column, and may have not been visible. 

'author action' We have replaced the term population by the term sample where
appropriate, rearranged the table format, and included the table in the manuscript
as Table 1.

'reviewer comment' The summary for seroprevalence should be consistent and
self-explanatory. E.g. 9 of 904 instead of percent, first round, second round (what
does it mean?)

'author response' The summary of seroprevalence estimates have been now presented in
a consistent way across all references.

'author action' The Table 1 has been amended accordingly

'reviewer comment' Line 113: I do not understand this sentence. What is Q1-Q3?
Quartiles for the minimum?

'author response' Q1 and Q3 are traditional mathematic expression for first and third
quartiles. Adding % makes it more explicit.

'author action' This sentence has been rewritten

'reviewer comment' Line 125: I suggest moving this to the study characteristics
section

'author response' We agree with this suggestion

'author action' Done

'reviewer comment' Discussion

Line 150: I do not understand this sentence

'author response' This sentence has been deleted

'author action' This sentence has been deleted

'reviewer comment' Line 155: this should be stated more clearly already in the
results in order to make this statement here.

'author response' Only twelve articles explicitely assessed asymptomatic
populations

'author action' This is now specified in the result section

'reviewer comment' Para 2: I think statements such as “Populations were not
differently affected….” are not allowed based on the design of your study. Even if
the majority of studies found no difference, this statement is not correct (if a
meta-regression would have been conducted, some statements in this direction could
have been made). Statements like most studies…found no…would be ok.

'author response' Thank you for this comment. 

'author action' The text has been amended.

to reviewers.docx
---

## [Decision Letter · Decision Letter 1]

28 Mar 2022

PONE-D-21-32219R1Seroprevalence and
SARS-CoV-2 invasion in general populations: a scoping review over the first year of
the pandemic.PLOS ONE

Dear Dr. METZGER,

Thank you for submitting your manuscript to PLOS ONE. After careful consideration, we
feel that it has merit but does not fully meet PLOS ONE’s publication criteria as it
currently stands. Therefore, we invite you to submit a revised version of the
manuscript that addresses the points raised during the review process.

Please submit your revised manuscript by May 12 2022 11:59PM. If you will need more
time than this to complete your revisions, please reply to this message or contact
the journal office at plosone@plos.org. When
you're ready to submit your revision, log on to https://www.editorialmanager.com/pone/ and select the 'Submissions
Needing Revision' folder to locate your manuscript file.

Please include the following items when submitting your revised
manuscript:A rebuttal letter that responds to each point raised by the academic
editor and reviewer(s). You should upload this letter as a separate file
labeled 'Response to Reviewers'.A marked-up copy of your manuscript that highlights changes made to the
original version. You should upload this as a separate file labeled
'Revised Manuscript with Track Changes'.An unmarked version of your revised paper without tracked changes. You
should upload this as a separate file labeled 'Manuscript'.If you would like to make changes to your financial disclosure,
please include your updated statement in your cover letter. Guidelines for
resubmitting your figure files are available below the reviewer comments at the end
of this letter.

We look forward to receiving your revised manuscript.

Kind regards,

Chandrabose Selvaraj, Ph.D.

Academic Editor

PLOS ONE

Journal Requirements:

Reviewers' comments:

Reviewer's Responses to Questions

6. Review Comments to the Author

Reviewer #1: Thank you for addressing the suggested revisions. I know it must have
taken a lot of work to include 71 additional studies so well done.

Five minor comments:

Please add to the limitations that you did not conduct a grey literature search and
the rationale for this.

Page 13 line 273 – comment regarding blood donor studies “completely fading away in
2020.” They didn’t. Many blood donor studies are still ongoing. Please remove that
statement or revise to indicate that they “decreased in frequency”.

Page 15 line 328: “which peaked at 69% in England”. I would remove this part of the
statement. This English study was likely of very poor quality and 69% is probably a
very biased estimate.

Page 15, Line 328-329 “As the geographical representation becomes broader with time,
the representation of the spread of the pandemic in the world population improves.”
I would remove this statement – I don’t understand what it means or what its purpose
is.

Although the manuscript is understandable overall, there are grammatical issues
throughout. It would be useful to have two colleagues that have not seen this work
proofread for grammar prior to final submission. PLOS One does not copy edit
manuscripts and I do not have the time to type out all the errors for correction. I
apologize for not having the time for this but it is the reality.

---

## [Author Response · Author response to Decision Letter 1]

14 Apr 2022

Reviewer #1

Requirement Thank you for addressing the suggested revisions. I know it must have
taken a lot of work to include 71 additional studies so well done.

'author response' We appreciate the nice comment, thank you

'author action' --

Requirement Five minor comments:

Please add to the limitations that you did not conduct a grey literature search and
the rationale for this

'author response' We have added this to the limitations 

.

'author action' This is now added line 225-228:

Second, we did not include a grey literature search because we did not know how to
search such literature in the particular context of this pandemic with so much
suspicion around non-peer–reviewed publications.

Requirement Page 13 line 273 – comment regarding blood donor studies “completely
fading away in 2020.” They didn’t. Many blood donor studies are still ongoing.
Please remove that statement or revise to indicate that they “decreased in
frequency”

'author response' We agree with this suggestion

'author action' This sentence has been rewritten line 204

Requirement Page 15 line 328: “which peaked at 69% in England”. I would remove this
part of the statement. This English study was likely of very poor quality and 69% is
probably a very biased estimate.

'author response' We did not assess the quality of the studies at at the time of
study selection nor during extraction; this criterion was not part of our exclusion
criteria. Other studies with lower prevalence may also be of poor quality and
therefore biased. This is probably a point to mention in the limitation. We
acknowledge that it is not relevant to mention it in the conclusion because it was
not fully the scope of the review.

'author action' As a limitation, we have added a statement warning the reader that we
did not assess the methodological quality of the studies, lines 231-232, and removed
this part of the statement from the conclusion.

Requirement Page 15, Line 328-329 “As the geographical representation becomes broader
with time, the representation of the spread of the pandemic in the world population
improves.” I would remove this statement – I don’t understand what it means or what
its purpose is.

'author response' This sentence was not quite clear.

'author action' This sentence has been deleted

Requirement Although the manuscript is understandable overall, there are grammatical
issues throughout. It would be useful to have two colleagues that have not seen this
work proofread for grammar prior to final submission. PLOS One does not copy edit
manuscripts and I do not have the time to type out all the errors for correction. I
apologize for not having the time for this but it is the reality.

'author response' We are sorry for the grammatical errors and we requested a native
English-speaking medical editor to revise our manuscript.

'author action' The article has been edited for English language.

to Reviewers.docx
---

## [Decision Letter · Decision Letter 2]

16 May 2022

Seroprevalence and SARS-CoV-2 invasion in general populations: a scoping review over
the first year of the pandemic.

PONE-D-21-32219R2

Dear Dr. METZGER,

We’re pleased to inform you that your manuscript has been judged scientifically
suitable for publication and will be formally accepted for publication once it meets
all outstanding technical requirements.

Kind regards,

Chandrabose Selvaraj, Ph.D.

Academic Editor

PLOS ONE

Additional Editor Comments (optional):

Reviewers' comments:

Reviewer's Responses to Questions

**Comments to the Author**

1. If the authors have adequately addressed your comments raised in a previous round
of review and you feel that this manuscript is now acceptable for publication, you
may indicate that here to bypass the “Comments to the Author” section, enter your
conflict of interest statement in the “Confidential to Editor” section, and submit
your "Accept" recommendation.

Reviewer #1: All comments have been addressed

---

## [Editor Report · Acceptance letter]

15 Jun 2022

PONE-D-21-32219R2 

Seroprevalence and SARS-CoV-2 invasion in general populations: a scoping review over
the first year of the pandemic 

Dear Dr. Metzger:

I'm pleased to inform you that your manuscript has been deemed suitable for
publication in PLOS ONE. Congratulations! Your manuscript is now with our production
department. 

Kind regards, 

on behalf of

Dr. Chandrabose Selvaraj 

Academic Editor

PLOS ONE